# HNF1A is a novel oncogene that regulates human pancreatic cancer stem cell properties

Ethan V Abel[1,2], Masashi Goto[2], Brian Magnuson[2,3], Saji Abraham[2], Nikita Ramanathan[2], Emily Hotaling[2], Anthony A Alaniz[2], Chandan Kumar-Sinha[4], Michele L Dziubinski[1,2], Sumithra Urs[2], Lidong Wang[5,6], Jiaqi Shi[2,4], Meghna Waghray[2], Mats Ljungman[2,7], Howard C Crawford[1,2], Diane M Simeone[5,6,8]*

[1]Department of Molecular and Integrative Physiology, University of Michigan Health System, Ann Arbor, United States; [2]Translational Oncology Program, University of Michigan Health System, Ann Arbor, United States; [3]Department of Biostatistics, School of Public Health, University of Michigan Health System, Ann Arbor, United States; [4]Department of Pathology, University of Michigan Health System, Ann Arbor, United States; [5]Department of Surgery, New York University Langone Health, New York, United States; [6]Perlmutter Cancer Center, New York University Langone Health, New York, United states; [7]Department of Radiation Oncology, University of Michigan Health System, Ann Arbor, United States; [8]Department of Pathology, New York University Langone Health, New York, United States

*For correspondence:
diane.simeone@nyumc.org

Competing interests: The authors declare that no competing interests exist.

**Abstract** The biological properties of pancreatic cancer stem cells (PCSCs) remain incompletely defined and the central regulators are unknown. By bioinformatic analysis of a human PCSC-enriched gene signature, we identified the transcription factor HNF1A as a putative central regulator of PCSC function. Levels of HNF1A and its target genes were found to be elevated in PCSCs and tumorspheres, and depletion of HNF1A resulted in growth inhibition, apoptosis, impaired tumorsphere formation, decreased PCSC marker expression, and downregulation of *POU5F1/OCT4* expression. Conversely, HNF1A overexpression increased PCSC marker expression and tumorsphere formation in pancreatic cancer cells and drove pancreatic ductal adenocarcinoma (PDA) cell growth. Importantly, depletion of HNF1A in xenografts impaired tumor growth and depleted PCSC marker-positive cells in vivo. Finally, we established an HNF1A-dependent gene signature in PDA cells that significantly correlated with reduced survivability in patients. These findings identify HNF1A as a central transcriptional regulator of PCSC properties and novel oncogene in PDA.
DOI: https://doi.org/10.7554/eLife.33947.001

## Introduction

Pancreatic ductal adenocarcinoma (PDA) is projected to be the second leading cause of cancer deaths in the U.S. by 2020 (*Rahib et al., 2014*). The exceeding lethality of PDA is attributed to a complex of qualities frequent to the disease including early and aggressive metastasis and limited responsiveness to current standards of care. While both aspects are in-and-of-themselves multifaceted and can be partially attributed to factors such as the tumor microenvironment (*Olive et al., 2009*; *Provenzano et al., 2012*; *Waghray et al., 2016*) and the mutational profile of the tumor cells

**eLife digest** Pancreatic ductal adenocarcinoma is the most common form of pancreatic cancer. It is also one of the deadliest types of cancer: fewer than one in ten patients live for five years after being diagnosed with the disease. Several reasons can explain this poor outcome including that the cancer is often diagnosed late, when tumor cells have already spread, and that there are not many effective treatments for it.

Pancreatic tumors contain different types of cancer cells with different properties. Among these are the so-called pancreatic cancer stem cells. These aggressive cells produce copies of themselves, contributing to tumor growth and spread. They can also help tumors to resist chemotherapy and radiotherapy. New treatments that specifically target cancer stem cells could therefore prove important for treating pancreatic cancer.

It is still not clear what makes pancreatic cancer stem cells so aggressive, or how they differ from the rest of the cells in a tumor. Abel et al. therefore looked for proteins that were more abundant in human pancreatic cancer stem cells than in other, less aggressive cancer cells with the idea that these proteins are likely to be important for the behavior of the pancreatic cancer stem cells.

Abel et al. found that a protein called HNF1A is enriched in pancreatic cancer stem cells. Experimentally reducing the levels of HNF1A in cells taken from human pancreatic cancers caused the cells to grow less well and form smaller tumors when injected into the pancreases of mice. These tumors contained few cancer stem cells, suggesting that HNF1A is important for maintaining the stem cell state. Further experiments showed that HNF1A increases the amount of many other proteins inside cells, including one that controls the activity of normal stem cells.

Given the importance of HNF1A to pancreatic cancer stem cells, finding ways to prevent this protein from working could lead to new treatments for pancreatic cancer. At the moment there are no drugs that target HNF1A. Further research is therefore needed to develop new drugs that work against HNF1A or one of the other proteins that it affects.

DOI: https://doi.org/10.7554/eLife.33947.002

(*Yachida et al., 2012*), cancer stem cells (CSCs) have also been identified to contribute to promoting early metastasis and resistance to therapeutics (*Hermann et al., 2007*; *Li et al., 2011*).

CSCs, which were originally identified in leukemias (*Bonnet and Dick, 1997*; *Graham et al., 2002*), have been identified in a number of solid tumors including glioblastoma (*Singh et al., 2003*), pancreas (*Li et al., 2007*; *Hermann et al., 2007*) and colon (*O'Brien et al., 2007*). In these cases, CSCs have been characterized by the ability to establish disease in immunocompromised mice, to resist chemotherapeutics, the capability of both self-renewal and differentiation into the full complement of heterogeneous neoplastic cells that comprise the tumor, and the propensity to metastasize. In each case, CSCs are distinguished from other tumor cell types by the expression of various, sometimes divergent cell surface markers. Our lab was the first to identify pancreatic cancer stem cells (PCSCs), which were found to express the markers EPCAM (ESA), CD44, and CD24 (*Li et al., 2007*). In addition to these markers, CD133 (*Hermann et al., 2007*), CXCR4 (*Hermann et al., 2007*), c-MET (*Li et al., 2011*), aldehyde dehydrogenase 1 (ALDH1) (*Kim et al., 2011*), and autofluorescence (*Miranda-Lorenzo et al., 2014*) have all been proposed markers of PCSCs. In all cases, the identified cells are characterized by being able to form spheres of cells (tumorspheres) under non-adherent, serum-free conditions, as well as an increased ability to form tumors in mice compared to bulk tumor cells. While a number of markers have been identified for PCSCs, relatively little is known about the transcriptional platforms that govern their function and set them apart from the majority of bulk PDA cells. Transcriptional regulators such as NOTCH (*Wang et al., 2009*; *Abel et al., 2014*), BMI1 (*Proctor et al., 2013*), and SOX2 (*Herreros-Villanueva et al., 2013*) have been demonstrated to play roles in PCSCs, although these proteins are also critical for normal stem cell function in many tissues.

In this study, we sought to better understand the biological heterogeneity of PCSCs and their bulk cell counterparts in an effort to identify novel regulators of PCSCs in the context of low-passage, primary patient-derived PDA cells. Using microarray analysis and comparing primary PDA cell subpopulations with different levels tumorigenic potential and stem-cell-like function, we identified

hepatocyte nuclear factor 1-alpha (HNF1A), an endoderm-restricted transcription factor, as a key regulator of the PCSC state. Supporting this hypothesis, depletion of HNF1A resulted in a loss of PCSC marker expression and functionality both in vitro and in vivo. Additionally, ectopic expression of HNF1A augmented PCSC properties in PDA cells and enhanced growth and anchorage-independence in normal pancreatic cell lines. Mechanistically, we found that HNF1A directly regulates transcription of the stem cell transcription factor *POU5F1/OCT4*, which is necessary for stemness in PCSCs. Based on these data, we postulate a novel pro-oncogenic function for HNF1A through its maintenance of the pancreatic cancer stem cell properties.

## Results

### An HNF1A gene signature dominates a PCSC gene signature

A transcriptional profile of PCSCs has yet to be established, and we hypothesized that such a profile would contain key regulators of the PCSC state. To pursue this hypothesis, we utilized a series of low-passage, patient-derived PDA cell lines to isolate PCSC-enriching and non-enriching subpopulations for comparative analysis. Using two of our previously described PCSC surface markers, CD44 and EPCAM (*Li et al., 2007*), we found that low-passage PDA cells generally formed three subpopulations (abbreviated P herein) based on surface staining: CD44$^{High}$/EPCAM$^{Low}$ (P1), CD44$^{High}$/EPCAM$^{High}$ (P2), or CD44$^{Low}$/EPCAM$^{High}$ (P3) (*Figure 1A*). Similar expression patterns were observed in 10 primary tumor samples (data not shown). Additionally, a CD44$^{Low}$/EPCAM$^{Low}$ subpopulation was observed in five samples (data not shown), consistent with our previous data (*Li et al., 2007*). Using previously described measures of PCSC function (*Li et al., 2007*; *Li et al., 2011*), including co-expression of the PCSC marker CD24 (*Figure 1—figure supplement 1A*), the abilities for isolated subpopulations to reestablish heterogeneous CD44 and EPCAM surface expression (*Figure 1B*), to form tumorspheres under non-adherent/serum-free culture conditions (*Figure 1C,D*), and to initiate tumors in immune-deficient mice (*Supplementary file 1*), we found that P2 cells showed greater enrichment for cells with PCSC properties than their P1 and P3 counterparts.

Using two primary PDA lines (NY8 and NY15), P1, P2, and P3 PDA cells were sorted by flow cytometry, prepped immediately for mRNA, and analyzed by Affymetrix GeneChip microarray and validated by quantitative RT-PCR. We found that P2 cells from both lines exhibited a signature of 50 genes that was upregulated (>1.5 fold) relative to both P1 and P3 cell counterparts (*Figure 1E*). To further refine this gene cohort, we utilized oPOSSUM (*Kwon et al., 2012*), a web-based system to detect overrepresented transcription-factor-binding sites in gene sets. Interestingly, HNF1A, a P2 cohort gene itself (*Figure 1E,F*), had predicted binding sites in the ±5000 regions (from start of transcription) of 17/50 of the enriched genes, and due to its stringent consensus sequence (DGTTAAT-NATTAAC) was the most highly ranked common transcription factor by Z-score (17.895). Of these 50 genes, HNF1A is known to positively regulate cohort genes *HNF4A* (*Boj et al., 2001*), *NR5A2* (*Molero et al., 2012*), *CDH17* (*Zhu et al., 2010*), *IGFBP1* (*Babajko et al., 1993*; *Powell and Suwanichkul, 1993*), and *DPP4* (*Gu et al., 2008*). Interestingly, genome-wide association (GWA) studies have recently identified certain single nucleotide polymorphisms (SNPs) in the HNF1A locus as risk factors for developing PDA (*Pierce and Ahsan, 2011*; *Li et al., 2012*; *Wei et al., 2012*), although the mechanism by which these SNPs exert their influence is currently unknown. Similarly, SNPs in the HNF1A target NR5A2 are also associated with the development of PDA (*Petersen et al., 2010*; *Rizzato et al., 2011*), further implicating a role for the HNF1A-transcriptional network in PDA. To further support the enrichment of HNF1A in PCSCs, sorted cells were western blotted for HNF1A and HNF1A-target proteins, CDH17 and DPP4. These proteins were found to be elevated in P2 cell lysates compared to other subpopulations (*Figure 1—figure supplement 1B*), in agreement with their transcript levels. CSCs are enriched in cancer cell populations grown under low-attachment tumorsphere (S) conditions compared to cells grown in adherent (A) conditions. In keeping with this observation, we found protein levels of HNF1A and CDH17 elevated in multiple PDA lines cultured under tumorsphere conditions (*Figure 1—figure supplement 2A and C*). Using a GFP-based reporter driven by eight tandem copies of the HNF1A consensus sequence GGTTAATGATTAACC (*Figure 1—figure supplement 2B*), we found GFP expression was elevated in NY5, NY8, and NY15 cells grown under tumorsphere (S) compared to adherent conditions (A) (*Figure 1—figure*

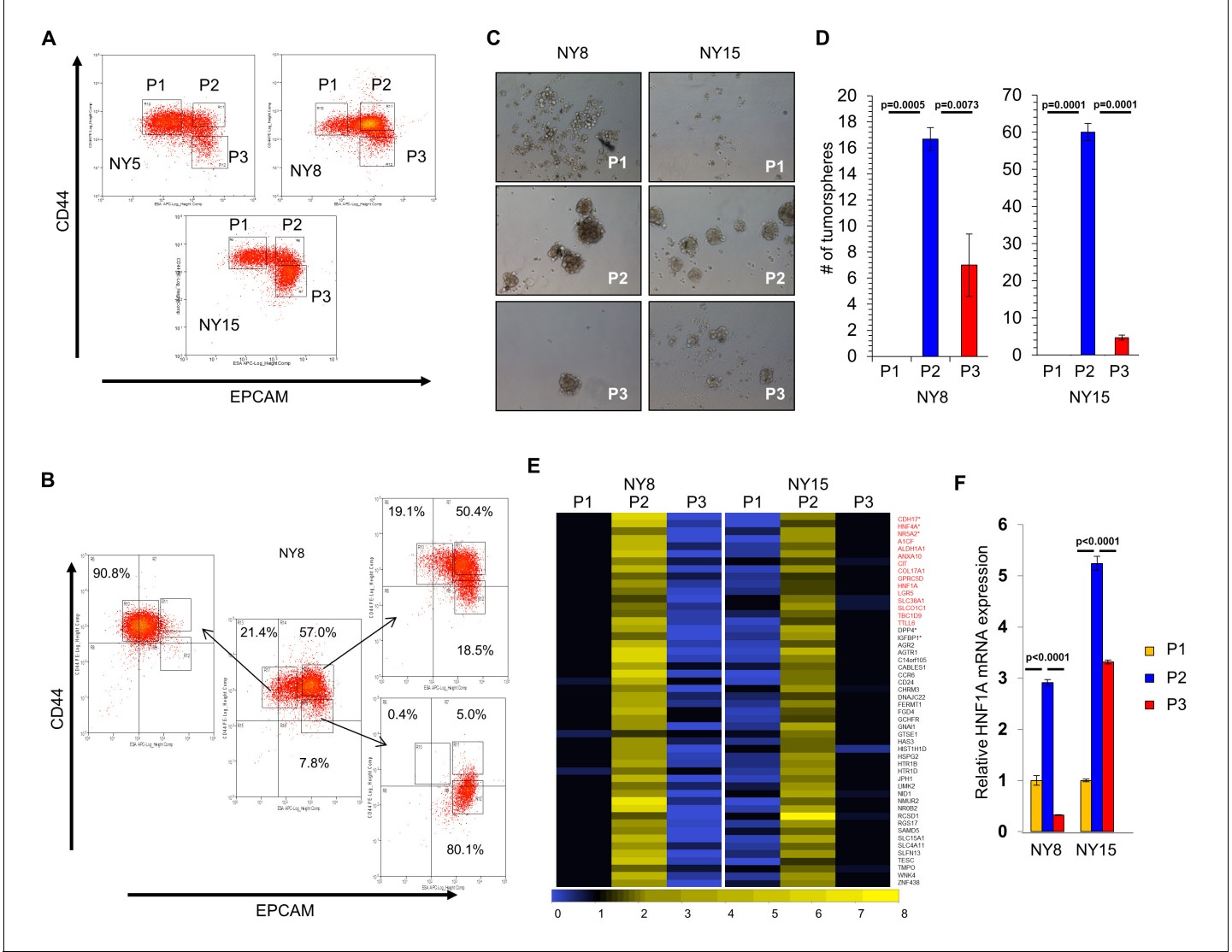

**Figure 1.** HNF1A-signature dominates pancreatic CSCs. (**A**) Flow cytometry analysis of CD44 and EPCAM surface expression of three primary PDA samples. (**B**) CD44$^{High}$/EPCAM$^{Low}$ (P1), CD44$^{High}$/EPCAM$^{High}$ (P2) and CD44$^{Low}$/EPCAM$^{High}$ (P3) NY8 cells were isolated by FACS and grown in culture for 17 days, followed by flow cytometry for analysis for CD44 and EPCAM expression. (**C, D**) Isolated subpopulations were grown in tumorsphere media on non-adherent plates (500 cells/well) for 6 days. Representative images of resultant tumorspheres (100X magnification) are shown in (**C**), while quantitation of spheres (n = 3) is shown in (**D**). Statistical difference was determined by one-way ANOVA with Tukey's multiple comparisons test. (**E**) Heat map representing relative fold differences in qRT-PCR expression of 50 cancer stem-cell-enriched genes in NY8 and NY15 cells. Per-gene values are relative to P1 or P3, whichever is higher. Gene names in red text indicate predicted HNF1A targets and asterisks (*) indicate known HNF1A targets. P1: CD44$^{High}$/EPCAM$^{Low}$, P2: CD44$^{High}$/EPCAM$^{High}$, P3: CD44$^{Low}$/EPCAM$^{High}$. For all genes, expression levels were normalized to an *ACTB* mRNA control, n = 3. Only genes with a significant (p<0.05) increase in P2 over both P1 and P3 subpopulations are shown, with statistical difference determined by one-way ANOVA with Tukey's multiple comparisons test. (**F**) qRT-PCR analysis of *HNF1A* mRNA expression, normalized to an *ACTB* mRNA control, from different primary PDA subpopulations (n = 3). Statistical difference was determined by one-way ANOVA with Tukey's multiple comparisons test. Related data can be found in *Figure 1—figure supplements 1* and *2*.

DOI: https://doi.org/10.7554/eLife.33947.003

The following source data and figure supplements are available for figure 1:

**Source data 1.** Quantitation of tumorspheres, P2 subpopulation-enriched transcripts, and HNF1A mRNA.

DOI: https://doi.org/10.7554/eLife.33947.006

**Source data 2.** Quantitation of GFP expression in adherent cells and tumorspheres.

DOI: https://doi.org/10.7554/eLife.33947.007

**Figure supplement 1.** Cancer stem cell properties of PDA cell subpopulations.

DOI: https://doi.org/10.7554/eLife.33947.004

*Figure 1 continued on next page*

*Figure 1 continued*

**Figure supplement 2.** HNF1A is elevated in tumorspheres.
DOI: https://doi.org/10.7554/eLife.33947.005

*supplement 2C*). This construct showed excellent dependence on HNF1A expression as targeting HNF1A with an HNF1A-specific siRNA ablated expression of both the ectopic GFP and endogenous CDH17 (*Figure 1—figure supplement 2D*). Lastly, we found the frequency of GFP-positive cells increased in cells grown in suspension (*Figure 1—figure supplement 2E*), with GFP expression being highest in the P2 subpopulation of NY15 cells (*Figure 1—figure supplement 2F*). Based on our gene expression and tumorsphere data, we hypothesized that HNF1A is a central regulator of CSC function.

## HNF1A is a critical regulator of CSC properties in PDA cells

Consistent with our hypothesis that HNF1A may be an integral component of PDA biology we observed higher levels of HNF1A protein and transcripts in PDA cells compared to non-transformed immortalized pancreatic cell lines HPNE (N) and HPDE (D) (*Figure 2A*; *Figure 2—figure supplement 1A*). Immunostaining of a PDA tissue microarray showed HNF1A expression to be significantly elevated ($p < 0.0001$) in PDA neoplastic ducts (n = 41) compared to normal pancreatic ducts (n = 18) (*Figure 2—figure supplement 1B,C*). To examine the role of HNF1A in PDA cells, we depleted the protein with two distinct siRNAs (*Figure 2B*). Knockdown of HNF1A resulted in reduced cell numbers in multiple primary PDA lines (*Figure 2C*). To determine whether the apparent loss in cell number was due to apoptotic cell death, we performed annexin V/DAPI staining on control and HNF1A-depleted NY5, NY8, and NY15 cells. In all cases, knockdown of HNF1A resulted in a significant ($p < 0.05$) increase in apoptotic cells, while not affecting necrotic cell numbers (*Figure 2D*, data not shown). Furthermore, increased cleavage of caspases 3, 6, 7, and 9 was observed in cells depleted of HNF1A (*Figure 2E*), indicating apoptotic cell death. These data indicate that HNF1A is important for PDA cell growth and survival.

Next, we pursued whether depletion of HNF1A impacted PDA subpopulation distribution. Consistent with a central role in maintaining heterogeneous EPCAM and CD44 expression, we observed a change in P2 in all cell lines (*Figure 3A*, *Figure 3—figure supplement 1A*) with a concomitant increase in the P3 population (*Figure 3—figure supplement 1A,C*). NY8 cells showed a loss in the P1 population as well (*Figure 3—figure supplement 1A,B*). Collectively, these results support a role for HNF1A in maintaining cellular heterogeneity, with the most dramatic change being the consistent loss of the PCSC population. In addition to changes in CD44 and EPCAM surface expression, we also observed a marked decrease in CD24 surface expression (*Figure 3B*, *Figure 3—figure supplement 1D*) and mRNA levels (data not shown) in multiple PDA lines; suggesting that loss of HNF1A depletes the CSC compartment. To assess functional consequences of HNF1A-depletion on the PCSC compartment, cells (NY5, NY8, NY15) expressing HNF1A shRNAs were grown under tumorsphere-promoting conditions. These shRNAs effectively depleted HNF1A as well as CDH17 (*Figure 3C*), indicating downstream signaling inhibition. Consistent with a role in PCSC function, HNF1A knockdown showed a marked reduction in tumorsphere formation ($p < 0.05$) (*Figure 3D,E*; *Figure 3—figure supplement 1E*).

## HNF1A exhibits oncogenic properties in pancreatic cells

We next sought to determine whether CSC properties could be augmented by ectopic expression of HNF1A in PDA cells. For these studies, we selected PDA lines with high (NY15), medium (NY8), and low (NY53) expression of HNF1A (*Figure 2A*) to determine if additional HNF1A expression could bolster PCSC properties under different cellular contexts. Using doxycycline-inducible expression of HNF1A (*Figure 4A,B*), we noted increased expression of CD24, CD44, and EPCAM in multiple primary PDA lines (*Figure 4B–D*, data not shown), indicating that ectopic HNF1A can increase PCSC marker expression in PDA cells. Additionally, we found that HNF1A-expressing cells formed ~2.5 fold more tumorspheres than their counterparts (*Figure 4E*) in all PDA cells tested. Taken together, these data indicate that ectopic HNF1A can promote PCSC properties, even in the presence of higher endogenous expression (i.e. NY15).

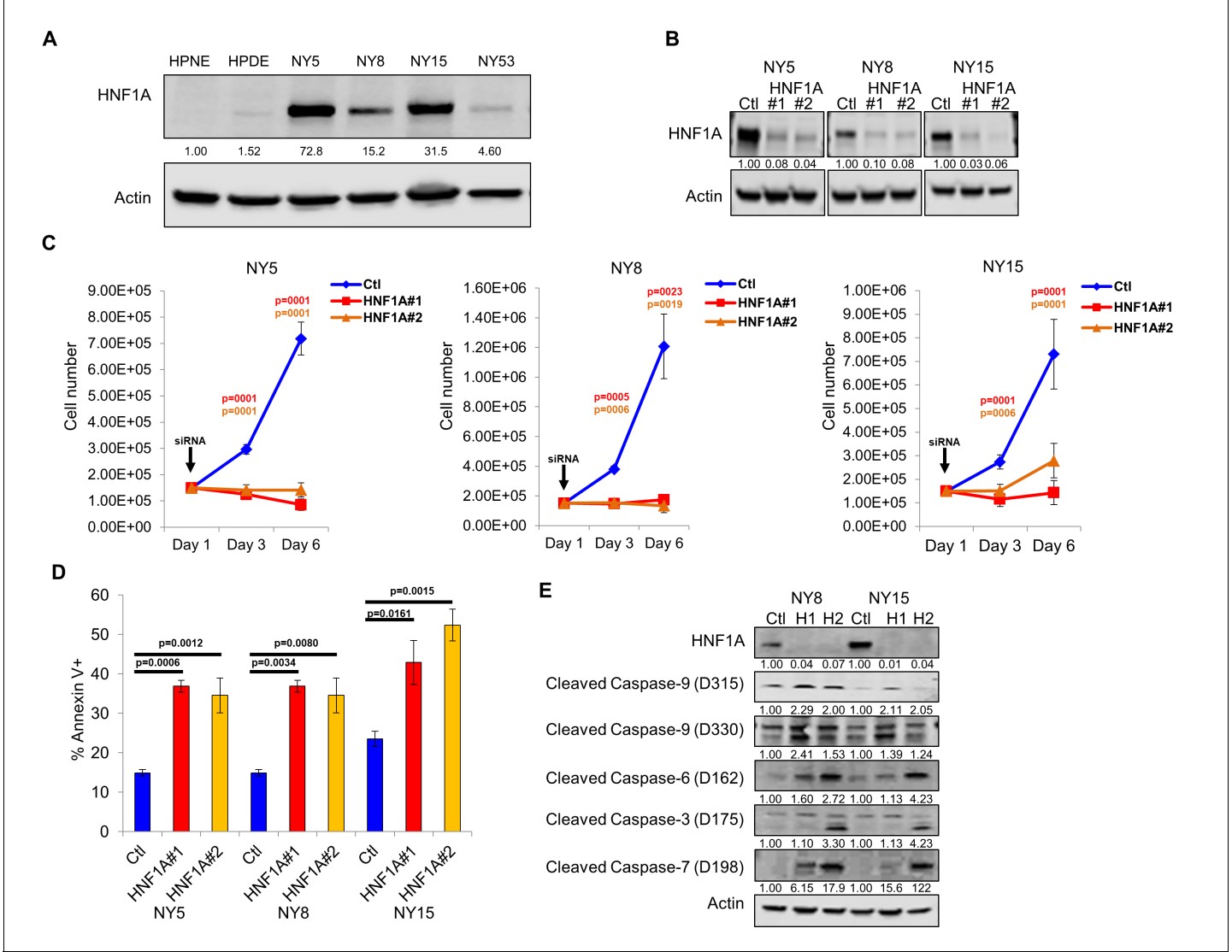

**Figure 2.** Knockdown of HNF1A in primary PDA cells inhibits growth in vitro. (**A**) Western blot analysis of HNF1A expression in a panel of primary PDA lines compared to immortalized pancreatic ductal cell line HPNE and HPDE. Quantitation of HNF1A protein is indicated below the respective blots. (**B**) Western blot of NY5, NY8, and NY15 cells transfected with non-targeting (Ctl) or HNF1A-targeting siRNA for 3 days, showing effective depletion of HNF1A protein by RNAi. Quantitation of HNF1A protein is indicated below the respective blots. (**C**) $1.5 \times 10^5$ PDA cells were transfected with control (Ctl) or HNF1A-targeting siRNA (Day 1). Cells were collected and manually counted 3 and 6 days after transfection (n = 3). Statistical difference was determined by one-way ANOVA with Dunnett's multiple comparisons test. Red and green p values indicate Ctl vs. HNF1A#1 or #2, respectively. (**D**) Annexin V staining was performed on NY5, NY8, and NY15 cells transfected with control (Ctl) or HNF1A-targeting siRNA (**H1, H2**) for 3 days. The amount of apoptotic (annexin V+) cells are quantitated (n = 4). Statistical difference was determined by one-way ANOVA with Dunnett's multiple comparisons test, with p values relative to the control siRNA group indicated. (**E**) Western blot analysis of cleaved caspases in NY8 and NY15 cells following HNF1A-knockdown (3 days). Actin serves as a loading control. Quantitation of proteins is indicated below the respective blots. Related data can be found in *Figure 2—figure supplement 1*.

DOI: https://doi.org/10.7554/eLife.33947.008

The following source data and figure supplement are available for figure 2:

**Source data 1.** Quantitation of PDA cell growth and apoptosis following HNF1A knockdown.
DOI: https://doi.org/10.7554/eLife.33947.010
**Source data 2.** Quantitative PCR analysis of HNF1A mRNA in PDA cells and histology score of HNF1A staining in normal and neoplastic ducts..
DOI: https://doi.org/10.7554/eLife.33947.011
**Figure supplement 1.** HNF1A expression in PDA cells and patient samples.
DOI: https://doi.org/10.7554/eLife.33947.009

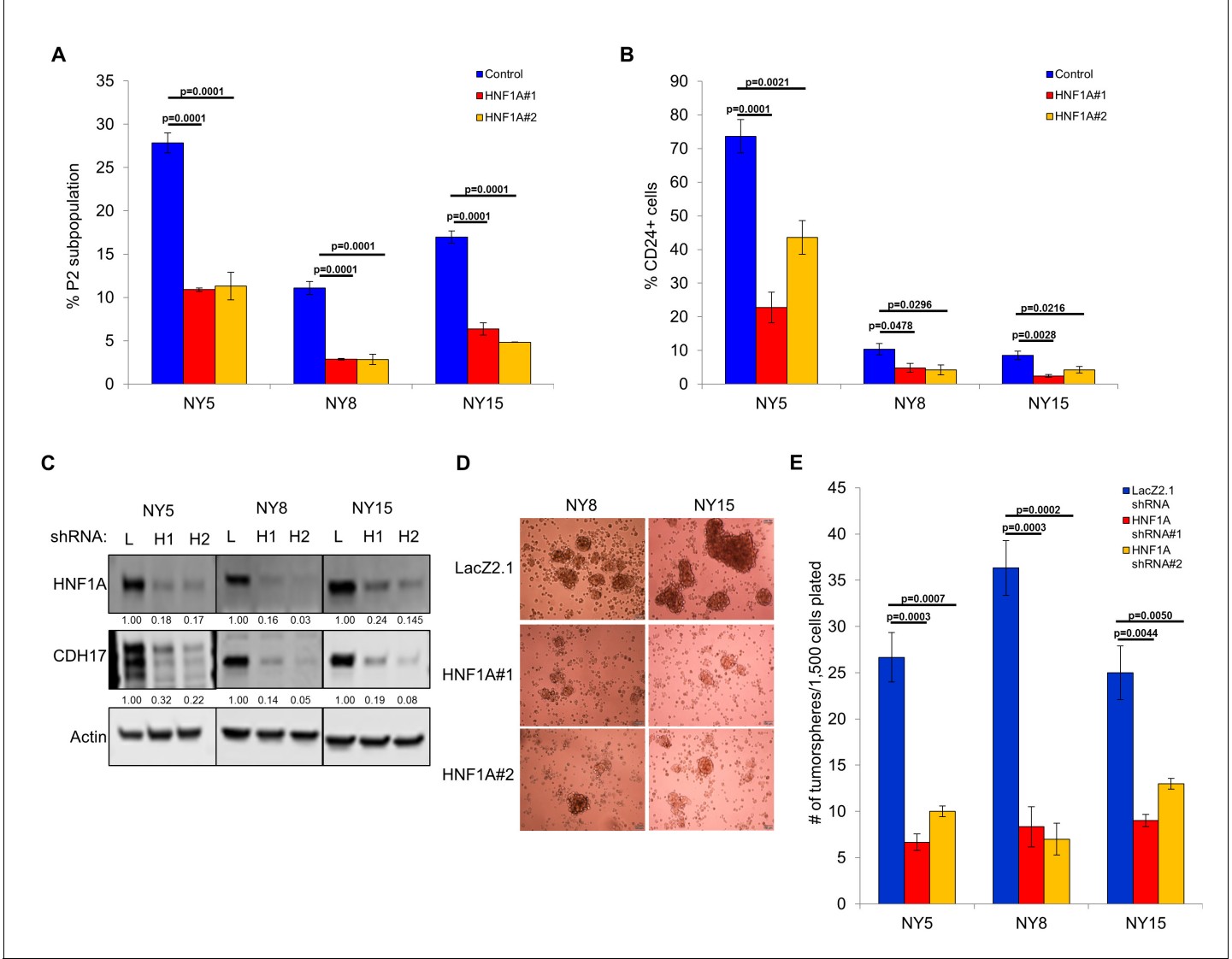

**Figure 3.** Knockdown of HNF1A depletes CSC numbers. (**A**) Multiple PDA cells were transfected with HNF1A-targeting siRNA or non-targeting control siRNA for 6 days. Surface expression of CD44 and EPCAM was measured by flow cytometry, and the percentage of CD44$^{High}$/EPCAM$^{High}$ (P2) cells are represented (mean ± SEM, n = 3). Statistical difference was determined by one-way ANOVA with Dunnett's multiple comparisons test. (**B**) Quantitation of CD24 +cells in multiple primary PDA cells following HNF1A knockdown for 6 days, n = 4. Statistical difference was determined by one-way ANOVA with Dunnett's multiple comparisons test. (**C**) NY5, NY8, and NY15 cells expressing LacZ2.1 (**L**) or two distinct HNF1A-targeting shRNAs (H1 and H2) were lysed and western blotted for HNF1A, CDH17, and Actin, showing effective knockdown of HNF1A and downstream signaling (CDH17). Quantitation of proteins is indicated below the respective blots. (**D, E**) NY5, NY8, and NY15 cells expressing LacZ2.1 or HNF1A-targeting shRNAs were grown in tumorsphere media on non-adherent plates (1500 cells/well). The number of tumorspheres formed after 6 days were counted (n = 3). Representative images of spheres (100X magnification) are shown in (**F**) and in *Figure 3—figure supplement 1*, with quantitation in (**E**). Statistical difference was determined by one-way ANOVA with Dunnett's multiple comparisons test. Related data can be found in *Figure 3—figure supplement 1*.

DOI: https://doi.org/10.7554/eLife.33947.012

The following source data and figure supplement are available for figure 3:

**Source data 1.** Quantitation of the P2 subpopulation, CD24 expression, and tumorsphere formation following HNF1A knockdown.
DOI: https://doi.org/10.7554/eLife.33947.014

**Figure supplement 1.** Knockdown of HNF1A depletes CSC numbers and properties in vitro.
DOI: https://doi.org/10.7554/eLife.33947.013

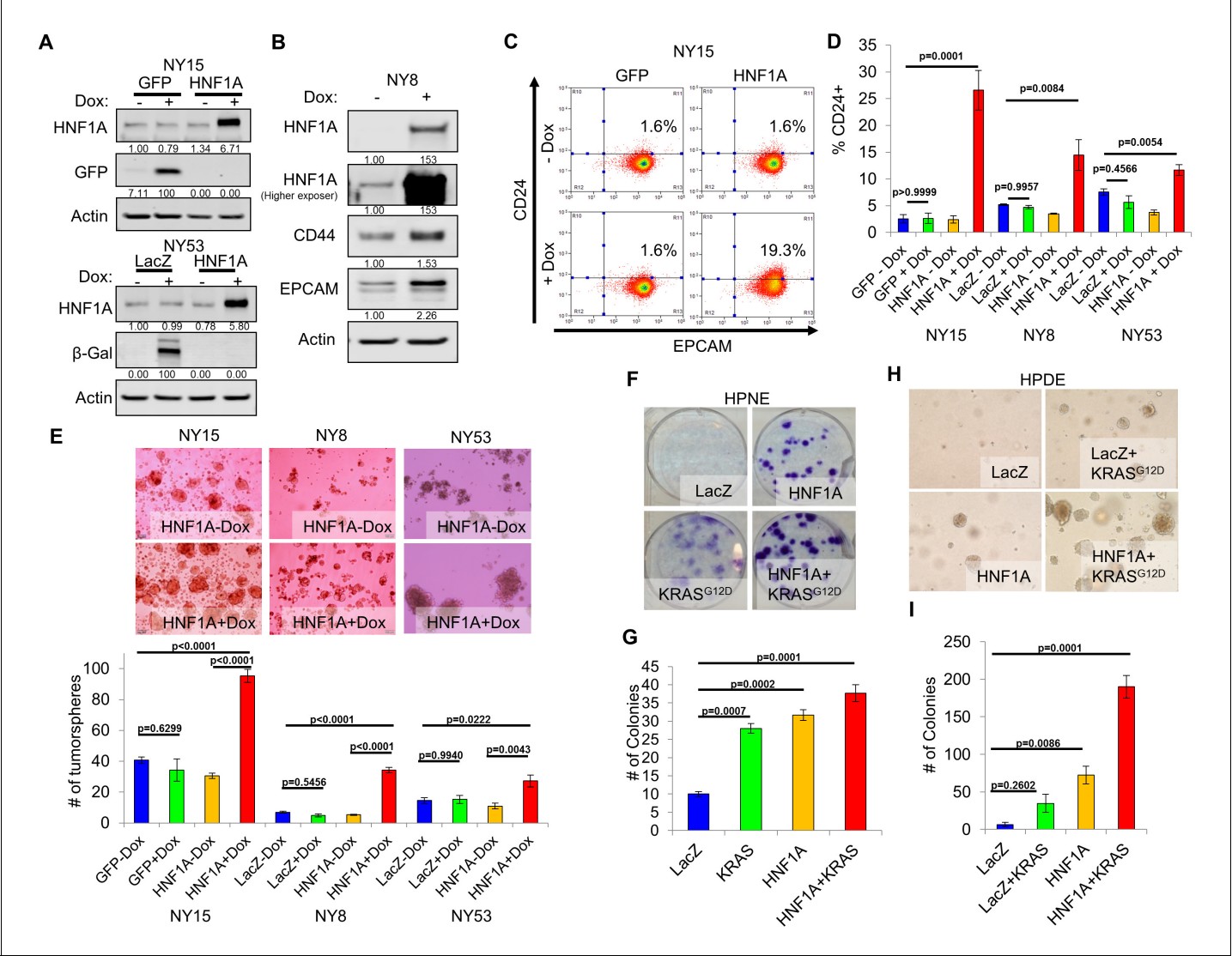

**Figure 4.** Overexpression of HNF1A promotes CSC properties in PDA cells and normal pancreatic cell lines. (**A**) NY15 and NY53 cells Western blotted for HNF1A and control gene induction following 48 hr ± doxycycline (Dox). Quantitation of proteins is indicated below the respective blots. (**B**) NY8 cells were treated 48 hr ± Dox to induce ectopic HNF1A. Lysates were western blotted for HNF1A, Actin, and PCSC markers EPCAM and CD44. Quantitation of proteins is indicated below the respective blots. (**C**) Representative surface expression of CD24 and EPCAM on NY15 cells expressing GFP or HNF1A. (**D**) Quantitation of CD24 +NY15 GFP and HNF1A cells and NY8 and NY53 LacZ and HNF1A cells by flow cytometry (n = 3). Statistical difference was determined by one-way ANOVA with Tukey's multiple comparisons test. (**E**) NY15 GFP and HNF1A, NY8 and NY53 LacZ and HNF1A cells were grown under sphere-forming conditions ± Dox. The number of tumorspheres formed after 7 days were quantitated (n = 4). Statistical difference was determined by one-way ANOVA with Tukey's multiple comparisons test. Representative images (100X magnification) of spheres are shown in the upper panels. (**F, G**) HPNE LacZ and HNF1A cells were plated at 200 cells/6 cm dish and treated ±Dox for 2 weeks, fixed, and stained with crystal violet (**F**). (**G**) Resultant colonies were quantitated (n = 3). Statistical difference was determined by one-way ANOVA with Dunnett's multiple comparisons test. (**H, I**) HPDE cells expressing inducible LacZ, LacZ with KRAS$^{G12D}$, HNF1A, or HNF1A with KRAS$^{G12D}$ were embedded in soft agar + Dox and monitored for signs of anchorage-independent growth for 21 days. (**H**) Representative images of resultant colonies (100X magnification) and (**I**) quantitation of colonies after 21 days (n = 3). Statistical difference was determined by one-way ANOVA with Dunnett's multiple comparisons test. Related data can be found in *Figure 4—figure supplement 1*.

DOI: https://doi.org/10.7554/eLife.33947.015

The following source data and figure supplement are available for figure 4:

**Source data 1.** Quantitation of CD24 expression and tumorsphere formation in PDA cells with HNF1A overexpression, and quantitation of colony formation in HPNE and HPDE cells expressing HNF1A and oncogenic KRAS.

DOI: https://doi.org/10.7554/eLife.33947.017

**Source data 2.** Quantitation of CD44+/CD24+ HPDE and HPNE cells overexpressing HNF1A.

*Figure 4 continued*

DOI: https://doi.org/10.7554/eLife.33947.018

**Figure supplement 1.** Overexpression of HNF1A and mutant KRAS in HPDE and HPNE cell.
DOI: https://doi.org/10.7554/eLife.33947.016

We next examined the effects of ectopic HNF1A expression in the non-tumorigenic pancreatic ductal cell lines HPDE and HPNE, which were devoid of endogenous HNF1A expression (*Figure 2A*). Doxycycline-inducible ectopic expression of HNF1A alone or in concert with ectopic KRAS$^{G12D}$ was readily achieved in HPDE cells (*Figure 4—figure supplement 1A*). Consistent with previous reports, KRAS$^{G12D}$-induced phosphorylation of both ERK1/2 and AKT in HPDE cells. Similar effects were seen in HPNE cells constitutively expressing HNF1A and KRAS$^{G12D}$ alone or in combination (*Figure 4—figure supplement 1A*). We then tested the impact the of HNF1A and/or KRAS$^{G12D}$ expression, either alone or in combination, on HPDE cell growth. Under normal growth conditions with serum, (LacZ) HPDE cells grew to confluency but did not form colonies, presumably due to contact-inhibition (*Figure 4—figure supplement 1B*). Expression of KRAS$^{G12D}$, however, resulted in colony formation, indicating a bypass of contact inhibition. HNF1A alone resulted in significantly increased colony formation, which was further enhanced by the additional expression of KRAS$^{G12D}$. Similar effects were seen in HPNE cells (data not shown). In clonogenicity assays, HNF1A-expressing HPNE cells formed similar numbers of colonies to control and KRAS$^{G12D}$-expressing cells (*Figure 4F, G*); however, HNF1A alone promoted enhanced colony size. HPDE cells failed to form colonies at clonal densities in the presence of serum. In addition to foci formation, anchorage-independent growth can indicate cellular transformation in vitro. When suspended in soft agar, control HPDE cells failed to grow over a 21-day period (*Figure 4H,I*). The addition of KRAS$^{G12D}$ alone did not significantly promote colony formation, consistent with its relatively weak transforming ability in HPDE cells. Interestingly, HNF1A alone resulted in numerous small colonies which in turn synergized with the expression of KRAS$^{G12D}$ in the form of numerous large colonies. Neither HNF1A nor KRAS$^{G12D}$ alone resulted in anchorage-independent growth in HPNE cells (data not shown). Lastly, we examined the effects of both transgenes on PCSC marker expression. Expression of HNF1A increased expression of EPCAM, CD44, and CD24 in HPDE cells (*Figure 4—figure supplement 1A,C*). Control HPNE cells lacked expression of both EPCAM and CD24, but expressed high levels of CD44. Expression of HNF1A was able to increase CD44 surface expression, while not changing EPCAM status (*Figure 4—figure supplement 1C*, data not shown). Remarkably, CD24 was potently induced upon HNF1A expression, with nearly 83% of HPNE cells expressing CD24 compared to 0.5% of LacZ-expressing control cells. These data would suggest that HNF1A possesses properties of an oncogene capable of cooperation with oncogenic KRAS.

## HNF1A is required for tumor growth and cancer stem cells properties in vivo

To determine whether HNF1A was necessary for tumorigenesis, we implanted two HNF1A-high primary lines (NY5 and NY15) expressing control or two HNF1A-targeting shRNAs orthotopically in the pancreas of NOD/SCID mice. HNF1A-depleted cells showed significantly reduced tumor growth compared to their control cohorts (p<0.05), (*Figure 5A,B*). Similar results were observed with HNF1A knockdown in subcutaneous xenografts of NY5 and NY15 cells (*Figure 5C*, *Figure 5—figure supplement 1A*). To determine whether inhibition of tumor growth was due to effects on the PCSC compartment, NY5 tumors were dissociated and analyzed by flow cytometry. Consistent with our in vitro findings, the EPCAM+/CD44+/CD24 +cell population was significantly reduced in HNF1A-depleted tumors (p<0.05) (*Figure 5D,E*). Importantly, western blot analysis of resultant tumor lysates confirmed that shRNAs remained effective at depleting HNF1A during the course of the experiment (*Figure 5—figure supplement 1B*).

## HNF1A regulates stemness through POU5F1/OCT4 expression

As a direct relationship between HNF1A and stem cell function has not been reported, we examined mRNA expression of central stemness regulators *MYC*, *SOX2*, *KLF4*, *NANOG*, and *POU5F1/OCT4* in HNF1A-depleted cells. Of these transcription factors, only *POU5F1/OCT4* mRNA showed consistent

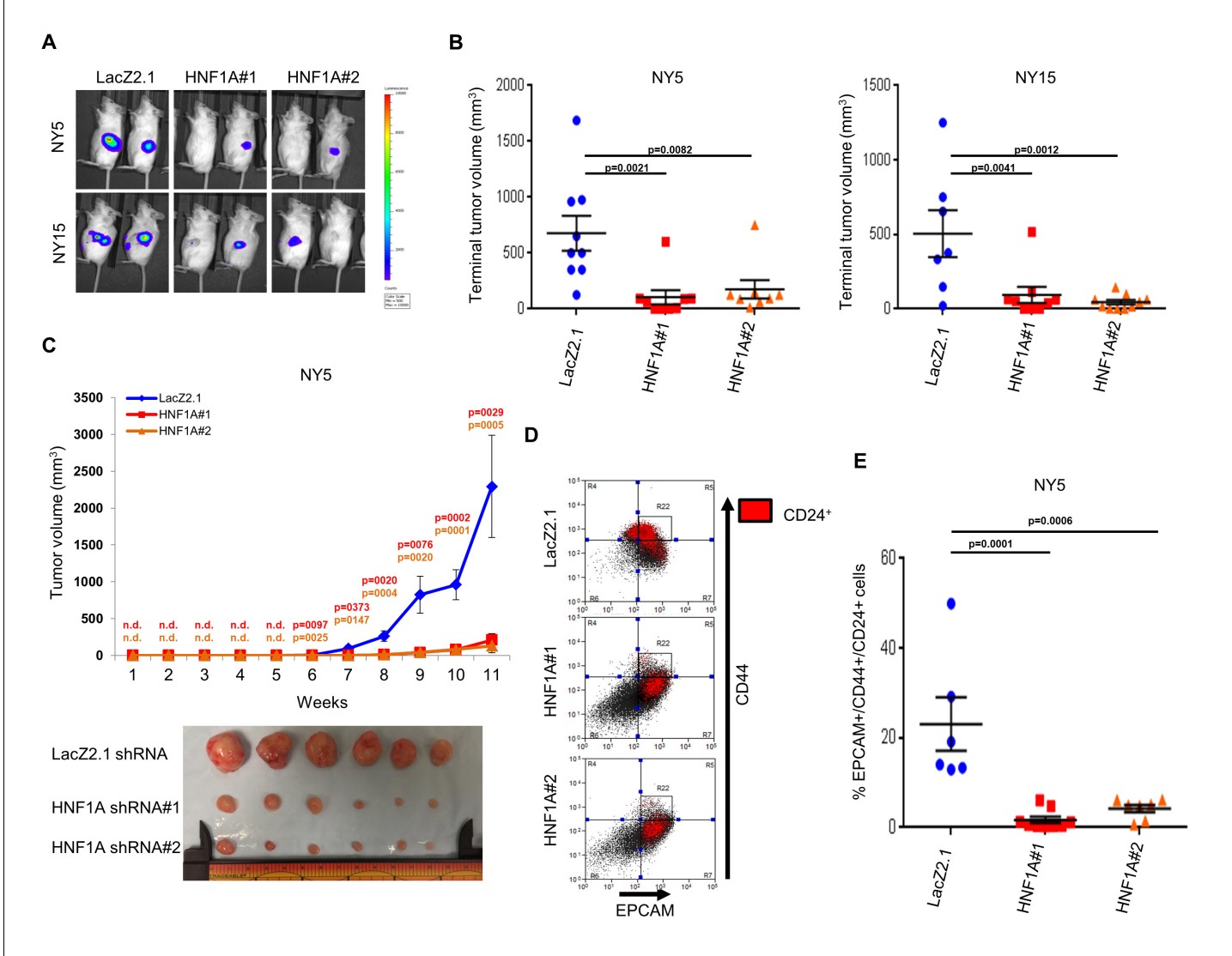

**Figure 5.** Knockdown of HNF1A impairs tumor growth and depletes CSCs in vivo. (**A, B**) 10,000 firefly luciferase-labeled NY5 and NY15 cells expressing control or HNF1A shRNAs were implanted orthotopically into the pancreata of NOD/SCID mice and monitored by IVIS imaging for 6 weeks (10 mice per group). Representative luminescence image of tumors prior to sacrifice is shown (**A**). Final tumor volumes determined during necropsy are quantitated in (**B**). Statistical difference was determined by one-way ANOVA with Dunnett's multiple comparisons test. (**C**) $10^3$ control or HNF1A-depleted NY5 cells were implanted subcutaneously in NOD/SCID mice (10 mice per shRNA/bilateral injections) for 11 weeks. Tumors were measured by caliper to determine tumor growth (**C**, upper panel). Statistical difference was determined by one-way ANOVA with Dunnett's multiple comparisons test. Red and orange p values indicate LacZ2.1 vs. HNF1A#1 or #2, respectively. 'n.d.' indicates that tumors were not detected. Representative tumors excised at sacrifice are shown (**C**, lower panel). (**D, E**) NY5 tumors from (**A**) were dissociated and stained for EPCAM, CD44, and CD24. Representative flow cytometry plots for recovered tumor cells are shown in (**D**), where the R22 gate denotes EPCAM$^{High}$/CD44$^{High}$ cells, and CD24 +cells are donated in red. Quantitation of EPCAM+/CD44+/CD24 +cells is shown in (**E**), n = 6 tumors each for shRNA. Statistical difference was determined by one-way ANOVA with Dunnett's multiple comparisons test. Related data can be found in *Figure 5—figure supplement 1*.

DOI: https://doi.org/10.7554/eLife.33947.019

The following source data and figure supplement are available for figure 5:

**Source data 1.** Quantitation of orthotopic and subcutaneous xenograft tumor volumes, and quantitation of PCSCs following HNF1A knockdown.
DOI: https://doi.org/10.7554/eLife.33947.021

**Source data 2.** Quantitation of subcutaneous xenograft tumor volumes following HNF1A knockdown.
DOI: https://doi.org/10.7554/eLife.33947.022

**Figure supplement 1.** Effects of HNF1A depletion on PDA xenograft biology.
DOI: https://doi.org/10.7554/eLife.33947.020

downregulation in multiple PDA cell lines in response to HNF1A knockdown (*Figure 6A*, data not shown). Similarly, we found that *POU5F1/OCT4* mRNA was upregulated in response to overexpression of HNF1A in both PDA cells and HPDE cells (*Figure 6B*), indicating regulation of *POU5F1/OCT4* expression by HNF1A in pancreatic-lineage cells. To determine whether *POU5F1/OCT4* mRNA was correlated with HNF1A expression, qRT-PCR was performed in 22 primary PDA lines as well as HPNE and HPDE cells. The Pearson correlation coefficient of *POU5F1/OCT4* mRNA was found to be significantly correlated (p=0.0094) with *HNF1A* mRNA levels (*Figure 6C*). Additionally, *POU5F1/OCT4* and *HNF1A* mRNA levels were correlated (Pearson's r = 0.406, p=8.9×10$^{-8}$) in patient tumors samples from The Cancer Genome Atlas (TCGA) dataset for PDA (PAAD cohort, data not shown), further supporting relationship between the two genes. Despite a strong association between *POU5F1/OCT4* and *HNF1A* mRNA levels, we did not observe a significant association between *POU5F1/OCT4* mRNA and any of the PDA subpopulations, indicating that factors other than HNF1A modulate the levels of *POU5F1/OCT4* mRNA in different PDA subpopulations (data not shown).

Previously published HNF1A chromatin immunoprecipitation sequencing (ChIP-seq) data performed in HepG2 cells by The Encyclopedia of DNA Elements (ENCODE) project (*Consortium and ENCODE Project Consortium, 2012*) identified a region of enrichment of HNF1A upstream of the POU5F1/PSORS1C3 gene loci proximal to recently identified retrotransposon long terminal repeat (LTR)-rich region that can serve as a promoter for both genes (*Malakootian et al., 2017*). Additionally, enrichment of this LTR region by TATA-binding protein (TBP) and acetylated lysine 27 histone H3 supports the involvement of this region in the transcription of *POU5F1/OCT4*. Interestingly, this LTR promoter region contains three consensus half-sites for HNF1A (*Figure 6—figure supplement 1A*). To test whether HNF1A binds directly to these half-sites, ChIP-PCR was performed in NY5, NY8, and NY15 cells. Consistent with the ENCODE data we observed significant enrichment of two half-sites in NY5 and all three half-sites in NY8 and NY15 by HNF1A. By contrast, the canonical distal enhancer of POU5F1/OCT4 (*Yeom et al., 1996*), located 14-kbp downstream of the LTR promoter, and HNF1A non-target gene MYOD showed no significant enrichment by HNF1A (*Figure 6—figure supplement 1B*), demonstrating the specificity of enrichment observed. To validate the LTR promoter region as an HNF1A-responsive promoter region, a reporter construct was generated encompassing the three putative HNF1A half-sites (*Figure 6—figure supplement 1C*). Co-transfection of 293FT cells (which lack endogenous HNF1A) with the LTR reporter and an HNF1A-expression plasmid resulted in a 4.5-fold induction of Cypridina luciferase expression over LacZ-expression plasmid co-transfected cells (*Figure 6—figure supplement 1D*). Additionally, neither the cloning vector nor the canonical downstream promoter region of POU5F1/OCT4 showed responsiveness to HNF1A expression, supporting the POU5F1/OCT4 LTR promoter as the HNF1A-responsive promoter for the gene.

POU5F1/OCT4 has previously been shown to be elevated in PCSCs (*Miranda-Lorenzo et al., 2014*; *Luo et al., 2017*), although a functional role for the protein has not been demonstrated in this context. To determine if POU5F1/OCT4 regulation was a key event in HNF1A-dependent stemness, we targeted POU5F1/OCT4 with multiple siRNAs, either in combination or as single sequences. Depletion of POU5F1/OCT4 resulted in a pronounced inhibition of tumorsphere formation, comparable to HNF1A knockdown (*Figure 6D–H*). To determine whether changes in apoptosis or cell cycle were responsible for the loss of tumorsphere formation in response to POU5F1/OCT4 knockdown, we performed annexin V/DAPI staining and propidium iodide staining in NY8 cells following transfection with POU5F1/OCT4 siRNA. Consistent with its role as a regulator of stemness in normal and cancer stem cells (*Okita et al., 2007*; *Takahashi and Yamanaka, 2006*; *Lu et al., 2013*; *Kumar et al., 2012*; *Nishi et al., 2014*), we did not observe changes in either apoptosis or cell cycle in response to POU5F1/OCT4 knockdown (*Figure 6—figure supplement 2A,B*). Importantly, knockdown of either HNF1A or POU5F1/OCT4 had comparable effects on the protein levels of OCT4A (*Figure 6D*), the isoform responsible for imparting stemness (*Lee et al., 2006*). To determine whether expression of OCT4A was sufficient to rescue stemness of PDA cells depleted of HNF1A, NY8, and NY15 cells were transduced with OCT4A-expressing lentiviruses or vector controls and transfected with HNF1A siRNA. Consistent with our previous results, loss of HNF1A impaired tumorsphere formation in both lines expressing the vector control, however, this effect was rescued by the expression of OCT4A (*Figure 6I*, *Figure 6—figure supplement 2C,D*). These data indicate that HNF1A mediates stemness of PCSCs through direct transcriptional regulation of *POU5F1/OCT4*.

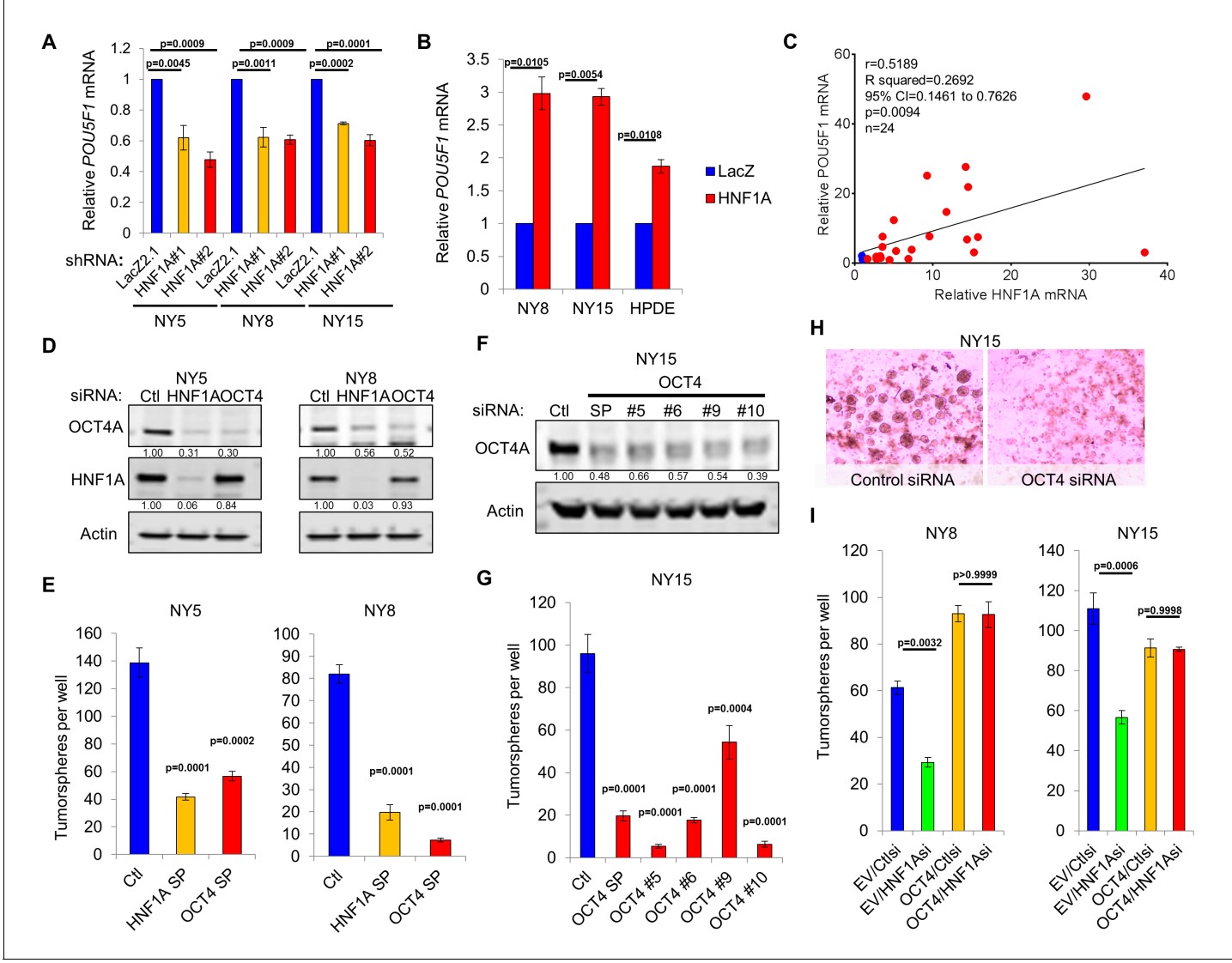

**Figure 6.** HNF1A regulates stemness through *POU5F1/OCT4* regulation. (**A**) qRT-PCR analysis of *POU5F1/OCT4* mRNA in NY5, NY8 and NY15 cells expressing control (LacZ2.1) or HNF1A shRNAs. *ACTB* was used as an internal control, n = 3. Statistical difference was determined by one-way ANOVA with Dunnett's multiple comparisons test. (**B**) LacZ or HNF1A was induced in NY8, NY15 or HPDE cells with doxycycline for 6 days. Levels of *POU5F1/OCT4* mRNA were measured by qRT-PCR with *ACTB* as an internal control, n = 3. Statistical difference was determined by unpaired t test with Welch's correction. (**C**) Pearson correlation coefficient of *POU5F1/OCT4* and *HNF1A* mRNA levels from NY PDA cells (n = 22, red) relative to HPNE and HPDE (blue) cells. (**D**) Western blot of OCT4A and HNF1A protein in NY5 and NY8 cells transfected with POU5F1/OCT4 (labeled OCT4) or HNF1A SMARTpool siRNA for 3 days. Quantitation of proteins is indicated below the respective blots. (**E**) NY5 and NY8 cells were transfected with HNF1A or POU5F1/OCT4 SMARTpool siRNA for 3 days and then grown in tumorsphere media on non-adherent plates (1500 cells/well). Spheres were quantitated 7 days later, n = 3. Statistical difference was determined by one-way ANOVA with Dunnett's multiple comparisons test. (**F–H**) NY15 cells were transfected with POU5F1/OCT4 SMARTpool (SP) siRNA or individual sequences for 3 days and either harvested to assess OCT4A knockdown by Western blot (**F**) or grown in tumorsphere media on non-adherent plates (1500 cells/well) (**G**). Spheres were quantitated 7 days later, n = 3. Statistical difference was determined by one-way ANOVA with Dunnett's multiple comparisons test. Representative spheres are shown in (**H**). (**I**) NY8 and NY15 cells transduced with OCT4A (labeled OCT4) or empty vector control (EV) were transiently transfected with control (Ctl) or HNF1A-targeting siRNA for 72 hr, and then grown in tumorsphere media on non-adherent plates (1500 cells/well). Spheres were quantitated 7 days later, n = 3. Statistical difference was determined by one-way ANOVA with Tukey's multiple comparisons test. Related data can be found in *Figure 6—figure supplements 1* and *2*.

DOI: https://doi.org/10.7554/eLife.33947.023

The following source data and figure supplements are available for figure 6:

*Figure 6 continued*

**Source data 1.** Quantitation of OCT4/POU5F1 mRNA following HNF1A knockdown and overexpression; relative HNF1A and OCT4/POU5F1 mRNA expressions in PDA cells; quantitation of tumorspheres following OCT4/POU5F1 knockdown; and quantitation of tumorsphere formation following OCT4A rescue.

DOI: https://doi.org/10.7554/eLife.33947.026

**Source data 2.** Quantitation of ChIP, CLuc activity, annexin V staining, PI staining, and tumorsphere formation.

DOI: https://doi.org/10.7554/eLife.33947.027

**Figure supplement 1.** HNF1A binds directly to and activates transcription from the POU5F1/OCT4 upstream LTR region.

DOI: https://doi.org/10.7554/eLife.33947.024

**Figure supplement 2.** Rescue of POU5F1/OCT4 expression in PDA cells.

DOI: https://doi.org/10.7554/eLife.33947.025

## HNF1A targets associated with poor survival in PDA patients

Lastly, we sought to gain insight into the transcriptional activity and genomic binding of HNF1A in PDA and determine whether its targets held prognostic information similar to other signatures in PDA (*Bailey et al., 2016*; *Collisson et al., 2011*). In order to identify transcriptional targets of HNF1A, we performed Bru-seq with control and HNF1A-depleted NY8 and NY15 cells (two replicates each of control shRNA and 2 HNF1A-targeting shRNAs per cell line). Bru-seq is a variation of RNA-seq which measures changes in nascent RNA levels (*bona fide* transcription rate) as opposed to steady-state mRNA changes measured by conventional RNA-seq and microarray (*Paulsen et al., 2013*). Differentially expressed genes were defined by adjusted p value<0.1 for at least one HNF1A-targeting shRNA and a mean expression level across samples (in RPKM) greater than 0.25. Of these differentially expressed genes, 243 HNF1A upregulated and 46 HNF1A downregulated were found to be in common between NY8 and NY15 (*Figure 7A*).

To assess genomic binding of HNF1A, we performed ChIP-seq using an HNF1A-specific antibody from NY8 and NY15 (two replicates each). ChIP-seq peaks were called using MACS (*Feng et al., 2012*) with the *a priori* assumption of narrow, transcription factor-like peaks. Input DNA was used to discern peaks from the background. Peaks were assigned to genes based on proximity and minimum mean expression level (0.25 RPKM) obtained from the Bru-seq data. Common peaks between NY8 and NY15 cells were defined as those peaks with overlap in at least one replicate of both cell lines. Genes were then classified as proximal, distal or neither, given the distance of the closest common peak to the transcription start site (proximal:±5 kb, distal:±100 kb, neither:>100 kb or no peak). The closest peak to a gene must also identify that gene as its closest gene, to discern among genes with nearby TSSs. 139/239 (57.2%) and 11/46 (23.9%) HNF1A upregulated/downregulated genes had HNF1A binding based on this criteria (*Figure 7B*), and supports the role of HNF1A as a transcriptional activator.

To further understand the regulatory role of HNF1A, we asked whether the HNF1A peaks overlapped with enhancer regions. The ENCODE combined segmentation model (a model for regulatory regions based on the ChromHMM and Segway models) was selected for this purpose (*Hoffman et al., 2013*; *Ernst and Kellis, 2012*; *Hoffman et al., 2012*). Of the six cell lines represented in this data set, it is not clear if any one best represents our PDA cell lines. We therefore extracted regions designated 'strong enhancer' (E) from all the cell lines and merged them into one set of enhancer regions. 72.7% of HNF1A-bound genes had peaks overlapping in at least one of these putative enhancer regions (*Consortium and ENCODE Project Consortium, 2012*) (*Figure 7B*), suggesting that HNF1A has significant interaction with regulatory regions.

A number of known HNF1A target genes exhibited HNF1A promoter-proximal binding and transcriptional responsiveness via Bru-seq/ChIP-seq, including *CDH17* (*Figure 7C*). Additionally, the PCSC marker *EPCAM* also showed HNF1A distal binding and transcriptional responsiveness, implicating HNF1A as a direct regulator of this gene. *CD24*, which showed decreased transcription in response to HNF1A loss, did not show direct binding, indicating an indirect mechanism of regulation (data not shown). *POU5F1/OCT4* transcription was found to decrease in both NY8 (34.3%) and NY15 (41.5%) cells, with weak enrichment of the LTR promoter region (data not shown), further supporting direct regulation of *POU5F1/OCT4* transcription by HNF1A. To determine whether POU5F1/OCT4 contributes to the deregulation of genes by HNF1A knockdown, we tested for over-representation of TF-binding motifs in the proximal promoter regions (±5 kbp from TSS) of HNF1A

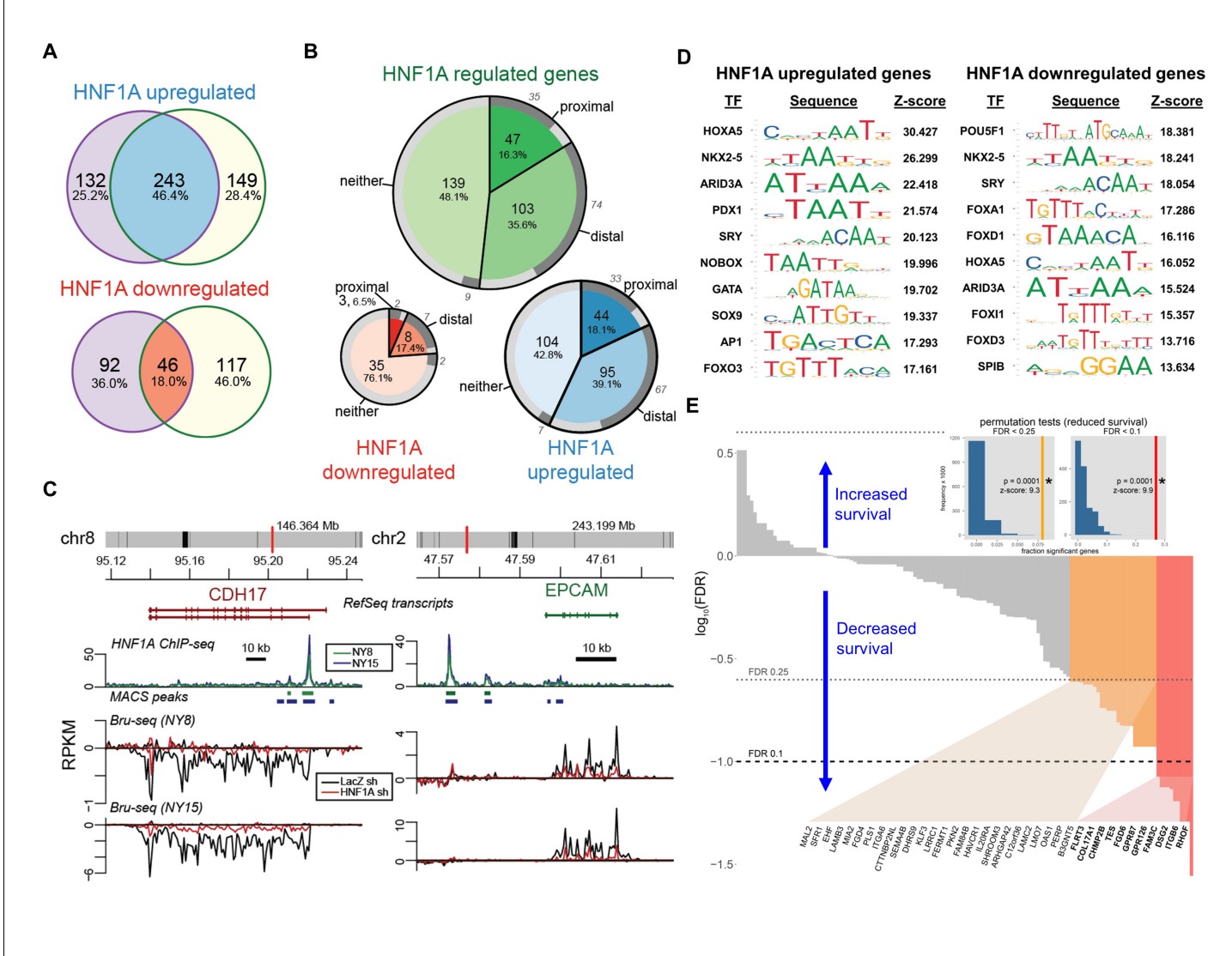

**Figure 7.** HNF1A regulates a transcriptional program associated with poor survival in PDA. (**A**) Venn diagrams illustrating overlapping genes with altered transcription (Bru-seq) following HNF1A knockdown in NY8 and NY15 cells. 'HNF1A upregulated' genes denote genes that were downregulated by HNF1A shRNAs, while 'HNF1A downregulated' genes were upregulated by HNF1A shRNAs. Cells expressed shRNAs constitutively for >14 days prior to Bru-seq analysis. (**B**) Proportion of HNF1A shRNA-downregulated genes identified in both NY8 and NY15 with HNF1A ChIP-seq peaks. Proximal peaks are ±5 kbp of the transcription start site (TSS) of a given gene and distal peaks are ±100 kbp of a TSS. Peaks are recognized only if they are closer to the TSS of a given gene than to other expressed genes. Peaks overlapping putative enhancer regions (ENCODE) are indicated in dark grey. (**C**) HNF1A ChIP-seq and HNF1A shRNA Bru-seq traces for the genes CDH17 and EPCAM in NY8 and NY15 cells. Traces represent normalized read coverage (in RPKM) across the indicated genomic ranges. MACS-identified ChIP peaks are represented by bars under the corresponding trace. (**D**) Transcription factor (TF) motif over-representation analysis of HNF1A upregulated and downregulated genes (±5 kbp of TSS). The top 10 over-represented TF motifs, ranked by z-score, are listed. (**E**) HNF1A upregulated and bound genes were ranked according to model significance and the direction of survival association using TCGA PDA patient data. Magnitude indicates significance ($\log_{10}$-transformed FDR-adjusted Wald p values for Cox PH models) and sign represents survival direction (determined by hazard ratio). Red bars indicate FDR < 0.1, orange bars indicate FDR < 0.25, and gray bars are not significant. The FDR thresholds are also indicated by dotted horizontal lines. Insets: each histogram represents the null distribution of a permutation test (N = 10,000) for fraction of genes significantly associated with reduced survival (the tests use FDR thresholds of 0.1 or 0.25, as indicated). Vertical lines represent values for the set of HNF1A target genes; red: FDR < 0.1 test; orange: FDR < 0.25 test. * - indicates significant p value estimates for the permutation tests. Related data found in *Figure 7—figure supplement 1*.

DOI: https://doi.org/10.7554/eLife.33947.028

The following figure supplement is available for figure 7:

**Figure supplement 1.** Association of HNF1A-responsive genes and survival in PDA.

DOI: https://doi.org/10.7554/eLife.33947.029

upregulated and downregulated genes using oPOSSUM. The POU5F1/OCT4 motif was the most significantly over-represented transcription factor motif in HNF1A downregulated genes (z-score = 18.381; 13/45 genes). The POU5F1/OCT4 motif was enriched in the HNF1A upregulated genes, though less highly ranked (rank #60; z-score = 2.104; 47/231 genes; *Supplementary file 2*). Of the predicted POU5F1/OCT4 targets, four have previously been identified (CACNA2D1, GATA2, SNAI1, and ZEB2) (Marsboom et al., *Li et al., 2010*; *Ben-Porath et al., 2008*). Additionally, other reported POU5F1/OCT4 targets (*Ben-Porath et al., 2008*) were identified among non-predicted targets, including the HNF1A upregulated genes KLF5 and ZHX2 and the HNF1A downregulated gene GJA1. These data demonstrate an overlap between HNF1A and POU5F1/OCT4 transcriptional networks.

Because CSC and oncogene gene signatures have been linked to prognosis in a variety of cancer types (*Bartholdy et al., 2014*; *Eppert et al., 2011*; *Glinsky et al., 2005*; *Merlos-Suárez et al., 2011*; *Rosenwald et al., 2003*), we asked if expression of HNF1-regulated genes was related to survival as a clinical outcome. The TCGA dataset for PDA (PAAD) consists of 178 tumor samples from different patients where both gene expression (RNA-seq) and clinical survival data was collected. Of these, we selected those tumors (n = 169) not identified as histologically neuroendocrine. For each gene, we generated a Cox proportional hazards survival model. We asked what fraction of genes in the HNF1A-responsive genes exhibited significance via Cox regression and whether they were associated with increased or reduced survival (hazard ratio <1 or>1, respectively). p Values were FDR-adjusted for multiple testing and two thresholds were explored. 13/237 (5.5%) of HNF1A upregulated genes were associated with reduced survival at FDR < 0.1 and 57/237 (24.1%) at FDR < 0.25, with only one gene associated with increased survival passing the FDR < 0.25 threshold (*Figure 7—figure supplement 1A*). For HNF1A upregulated and bound, we found a similar pattern; 11/137 (8.0%) genes associated with reduced survival and 37/137 (27.0%) genes at FDR < 0.25 and 0 genes passing the FDR < 0.25 threshold (*Figure 7E*). For HNF1A downregulated genes, 1/45 (2.2%) genes were significant at FDR < 0.25 only (*Figure 7—figure supplement 1B*). A background set of genes, defined as those genes expressed above a minimal threshold in the Bru-seq data and mappable to gene identifiers in the TCGA data (see Materials and methods, was selected for permutations testing). The permutation tests showed that HNF1A upregulated genes were significantly associated with poorer outcomes versus randomly selected genes (insets, *Figure 7E* and *Figure 7—figure supplement 1A*; see Materials and methods for details). These findings further support the oncogenic role for HNF1A in PDA as a direct regulator of a set of genes associated with poor patient survival.

## Discussion

In this study, we identified the transcription factor HNF1A as putative regulator of a PCSC gene signature. Functional studies revealed that HNF1A was not only central to the regulation of this gene signature, but also PCSC function. Depletion of HNF1A effectively inhibited PDA cell growth, tumorsphere formation, and tumor growth, with a loss of PCSC marker expression observed both in vitro and in vivo. Mechanistically, HNF1A appears to promote stemness through positive regulation of pluripotency factor POU5F1/OCT4. Finally, we found that expression of HNF1A upregulated genes significantly predicted poor survival outcomes in patients with PDA. These data point to a novel oncogenic role for HNF1A in pancreatic cancer, particularly in promoting PCSC properties.

A clear role for HNF1A in PDA has not previously been established. An early study of the putative oncogene FGFR4, frequently expressed in PDA (*Ohta et al., 1995*), is directly regulated by HNF1A through intronic binding sites (*Shah et al., 2002*). More recently, 73% of PDA samples were found to stain positive for HNF1A (*Kong et al., 2015*). A more direct role for HNF1A in PDA has been suggested by multiple GWA studies implicating certain SNPs in HNF1A as risk factors for the development of PDA (*Pierce and Ahsan, 2011*; *Wei et al., 2012*; *Li et al., 2012*). Nearly all the identified HNF1A SNPs are non-coding and relatively common (minor allele frequencies between 30 and 40%), suggesting these SNPs may serve as potential contributing rather than driving factors in pancreatic tumorigenesis. Interestingly, PDA-associated HNF1A SNPs rs7310409, rs1169300, and rs2464196 are also associated with both an elevated risk (1.5–2 fold) of developing lung cancer and elevated circulating C-reactive protein (CRP). A well-established direct target of HNF1A (*Toniatti et al., 1990*), CRP is downregulated in patients with inactivating mutations in HNF1A (*Thanabalasingham et al., 2011*). As several PDA-associated SNPs are associated with elevated

CRP, it is therefore possible that these SNPs augment the activity/expression of HNF1A rather than diminish it, as in the case of maturity-onset diabetes of the young 3 (MODY3) variants which reduce or abolish HNF1A expression or function. Still, a tumor suppressive role for HNF1A in PDA has also been proposed (*Hoskins et al., 2014*; *Luo et al., 2015*). In these studies, HNF1A was found to possess pro-apoptotic/anti-proliferative properties contrary to the data in this study. Differences in these results may be technical in nature (control cells in Luo et al. exhibited unusually high baseline apoptosis approaching 50%); however, it is also possible that the role of HNF1A may differ between different molecular subtypes of PDA (*Bailey et al., 2016*) or in a dynamic manner like fellow transcription factor PDX1 (*Roy et al., 2016*). Supporting the former, HNF1A expression has been proposed as a biomarker to distinguish between the exocrine/ADEX subtype (HNF1A high/KRT81 low) and the quasi-mesenchymal/squamous/basal-like subtype (HNF1A low/KRT81 high) (*Muckenhuber et al., 2018*; *Noll et al., 2016*), and supports previous observations that the quasi-mesenchymal/squamous/basal-like subtype is associated with poorer prognosis and drug resistance (*Bailey et al., 2016*; *Collisson et al., 2011*; *Moffitt et al., 2015*). Although these studies suggest that HNF1A expression may be highest in the exocrine/ADEX subtype of PDA, HNF1A function was not specifically examined. It is possible that like other pancreas-lineage transcription factors, such as PDX1 (*Roy et al., 2016*) and FOXA1 (*Roe et al., 2017*), HNF1A is associated with subtypes of PDA that retain elements of pancreatic identity (classical and exocrine/ADEX), but are nonetheless important maintenance of the disease. Interestingly, Noll et al. demonstrated that high expression of CYP3A5 in the exocrine/ADEX subtype mediates resistance to tyrosine kinase inhibitors and paclitaxel. Our work identifies CYP3A5 as a direct target of HNF1A, suggesting that HNF1A may play a direct role in drug resistance in PDA, and future studies should explore this possibility.

While we found an association between HNF1A upregulated genes and poor patient survival, we did not observe a significant association between HNF1A mRNA expression and survival (p=0.7017). As the promoters of HNF1A upregulated genes were enriched for transcription factor known to play roles in PDA including GATA (likely GATA5 or GATA6) (*Roe et al., 2017*; *Martinelli et al., 2017*; *Zhong et al., 2011*), PDX1 (*Roy et al., 2016*), and SOX9 (*Camaj et al., 2014*; *Kopp et al., 2012*; *Tsuda et al., 2018*), it is possible that HNF1A may work in concert with other transcription factors to elicit its full oncogenic function in PDA. A similar interaction between the transcription factors Foxa1 and Gata5 was recently described in driving metastasis in murine models of PDA (*Roe et al., 2017*).

Our data on HPDE and HPNE cells support a partially transforming capacity for HNF1A, wherein it overcomes contact-inhibition and anchorage-dependent growth. As cooperation with oncogenic KRAS was observed in these cells, it is feasible that HNF1A provides additional oncogenic input, possibly by altering the differentiation state of KRAS-mutant, precancerous pancreatic cells or by expanding the resident stem cell/cancer stem cell population. Indeed, expression of HNF1A alone was sufficient to increase CD24 expression/positivity in both HPDE and HPNE cells.

Typically a marker of endodermal differentiation, HNF1A has not previously been reported as necessary for normal or cancer stem cells. HNF1A plays a critical role in the normal functionality of the endocrine pancreas, with hereditary inactivating mutations in the gene and promoter region resulting in MODY3, an autosomal dominant form of diabetes resulting from β cell insufficiency. Additionally, murine knockout models recapitulate the diabetic phenotype seen in humans (*Lee et al., 1998*), with elegant transcriptomic work demonstrating a requirement for murine Hnf1a in β cell proliferation (*Servitja et al., 2009*). The role for HNF1A in the exocrine pancreas is less clear, and compared to islet and liver cells in the latter study, we only identified 11 overlapping HNF1A upregulated genes (*ANXA4*, *CEACAM1*, *CHKA*, *DPP4*, *HNF4A*, *HSD17B2*, *LGALS3*, *MTMR11*, *NR0B2*, *SLC16A5*, *TM4SF4*), suggesting distinct activity for HNF1A in PDA compared to either β cells or the liver. Regulation of *POU5F1/OCT4* transcription by HNF1A is an especially exciting finding, connecting HNF1A with a previously unidentified role in regulating stemness. Our study identifies a recently described LTR promoter region (*Malakootian et al., 2017*), upstream from the canonical POU5F1/OCT4 promoter, as a likely region of direct transcriptional regulation of POU5F1/OCT4 by HNF1A, supported by both ChIP and reporter assays (*Figure 6—figure supplement 1B, D*). As this promoter region, is not conserved between humans and rodents, it is possible the interaction between HNF1A and POU5F1/OCT4 is an acquisition of human evolution and may explain why POU5F1/OCT4 has not previously been identified as an HNF1A target. Interestingly, a recent study SPINK1-positive castrate resistant prostate cancer identified *POU5F1/OCT4* as part of a gastrointestinal gene signature present in SPINK1-positive prostate cancer and regulated by HNF1A

and its target gene HNF4G (*Shukla et al., 2017*). Consistent with our findings, this study showed downregulation/upregulation of *POU5F1/OCT4* mRNA in response to HNF1A knockdown/overexpression, respectively. While direct regulation of POU5F1/OCT4 and HNF1A was not explored in this study, it does support an association between these two transcription factors, not only in gastrointestinal cells, but other cancers as well. This could indicate a more general role for HNF1A in regulating stem cell properties in human cells in which it is normally expressed.

Given that HNF1A upregulated genes were found to be significantly associated with poor survival in patients with PDA, it is likely that multiple target genes contribute to HNF1A's oncogenic influence, and future studies should be done to assess the functions of these genes in PDA to ascertain their value as either potential biomarkers or therapeutic targets. Further studies are also needed in regards to HNF1A's role in the exocrine pancreas and whether its function is redirected during the development of PDA, particularly under the influence of oncogenic KRAS. Overall, this study further validates the importance of HNF1A to PDA while providing a novel and critical role for HNF1A in driving pancreatic cancer stem cell properties.

# Materials and methods

## Key resources table

| Reagent type (species) or resource | Designation | Source or reference | Identifiers | Additional information |
| --- | --- | --- | --- | --- |
| Gene (Human) | HNF1A | This paper | | Cloned from NY5 cDNA |
| Gene (Escherichia coli) | LacZ | Invitrogen | Originally from Catalog number: K499000 | Subcloned into pLentipuro3/TO/V5-DEST |
| Gene (Aequorea victoria) | PatGFP | This paper | | Variant of EGFP containing the following mutations: S31R, Y40N, S73A, F100S, N106T, Y146F, N150K, M154T, V164A, I168T, I172V, A207V |
| Gene (Human) | KRAS G12D | This paper | | Cloned from NY5 cDNA |
| Gene (Human) | POU5F1 (OCT4A) | Transomic Technologies | Catalog number: BC117435 | Subcloned into pLenti6.3 /UbC/V5-DEST |
| Gene (Escherichia coli) | LacZ2.1 shRNA | This paper | | Sequence: CACCAAATCGCTGATTT GTGTAGTCGTTCAAGAGACGACT ACACAAATCAGCGA |
| Gene (Human) | HNF1A shRNA#1 | This paper | | Sequence: CACCGCTAGTGGAGGA GTGCAATTTCAAGAGAATTGCACTC CTCCACTAGC |
| Gene (Human) | HNF1A shRNA#2 | This paper | | Sequence: CACCGTCCCTTAGTGA CAGTGTCTATTCAAGAGATAGA CACTGTCACTAAGGGAC |
| Gene (Escherichia coli) | IVS-TetR-P2A-Bsd | This paper | | IVS-TetR and Bsd were subcloned from pLenti6/TR (Invitrogen) with a P2A peptide linker added by PCR and Gibson Assembly |
| Gene (Aequorea victoria) | PatGFP-Luc2 | This paper | | PatGFP and Luc2 (Promega) were amplified by PCR and fused by Gibson Assembly |
| Strain, strain background (Mouse) | NOD.CB17-*Prkdc^scid*/J | The Jackson Laboratory | Catalog number: 001303; RRID: IMSR_JAX:001303 | |
| Cell line (Human) | HPDE | Craig Logsdon, MD Anderson | | |
| Cell line (Human) | HPNE | ATCC | Catalog number: ATCC CRL-4023; RRID:CVCL_C466 | |

*Continued on next page*

*Continued*

| Reagent type (species) or resource | Designation | Source or reference | Identifiers | Additional information |
|---|---|---|---|---|
| Cell line (Human) | Capan-2 | ATCC | Catalog number: ATCC HTB-80; RRID:CVCL_0026 | |
| Cell line (Human) | HPAF-II | ATCC | Catalog number: ATCC CRL-1997; RRID:CVCL_0313 | |
| Cell line (Human) | BxPC-3 | ATCC | Catalog number: ATCC CRL-1687; RRID:CVCL_0186 | |
| Cell line (Human) | AsPC-1 | ATCC | Catalog number: ATCC CRL-1682; RRID:CVCL_0152 | |
| Cell line (Human) | MiaPaCa-2 | ATCC | Catalog number: ATCC CRL-1420; RRID:CVCL_0428 | |
| Cell line (Human) | Panc-1 | ATCC | Catalog number: ATCC CRL-1469; RRID:CVCL_0480 | |
| Cell line (Human) | NY1 | This paper | | Low passage pancreatic adenocarcinoma patient primary cell line established from xenograft |
| Cell line (Human) | NY2 | This paper | | Low passage pancreatic adenocarcinoma patient primary cell line established from xenograft |
| Cell line (Human) | NY3 | This paper | | Low passage pancreatic adenocarcinoma patient primary cell line established from xenograft |
| Cell line (Human) | NY5 | This paper | | Low passage pancreatic adenocarcinoma patient primary cell line established from xenograft |
| Cell line (Human) | NY6 | This paper | | Low passage pancreatic adenocarcinoma patient primary cell line established from xenograft |
| Cell line (Human) | NY8 | This paper | | Low passage pancreatic adenocarcinoma patient primary cell line established from xenograft |
| Cell line (Human) | NY9 | This paper | | Low passage pancreatic adenocarcinoma patient primary cell line established from xenograft |
| Cell line (Human) | NY12 | This paper | | Low passage pancreatic adenocarcinoma patient primary cell line established from xenograft |
| Cell line (Human) | NY15 | This paper | | Low passage pancreatic adenocarcinoma patient primary cell line established from xenograft |
| Cell line (Human) | NY16 | This paper | | Low passage pancreatic adenocarcinoma patient primary cell line established from xenograft |

*Continued*

| Reagent type (species) or resource | Designation | Source or reference | Identifiers | Additional information |
|---|---|---|---|---|
| Cell line (Human) | NY17 | This paper | | Low passage pancreatic adenocarcinoma patient primary cell line established from xenograft |
| Cell line (Human) | NY19 | This paper | | Low passage pancreatic adenocarcinoma patient primary cell line established from xenograft |
| Cell line (Human) | NY28 | This paper | | Low passage pancreatic adenocarcinoma patient primary cell line established from xenograft |
| Cell line (Human) | NY32 | This paper | | Low passage pancreatic adenocarcinoma patient primary cell line established from xenograft |
| Cell line (Human) | NY53 | This paper | | Low passage pancreatic adenocarcinoma patient primary cell line established from xenograft |
| Cell line (Human) | 293FT | Invitrogen | Catalog number: R70007 | |
| Transfected construct (Gaussia) | pTK-GDLuc | This paper | | The Gaussia coding region of pTK-Gluc (New England Biolabs) was replaced with the Gaussia Dura coding region (Millipore) |
| Transfected construct (Cypridina) | pCLuc-Basic2 | New England Biolabs | Catalog number: N0317S | |
| Transfected construct (Cypridina) | pCLuc-Basic2/OCT4 LTR promoter | This paper | | 1.7 kbp OCT4 LTR promoter region from NY5 was subcloned into pCLuc-Basic2 |
| Transfected construct (Cypridina) | pCLuc-Basic2/OCT4 canonical promoter | This paper/Addgene | Originally from Catalog number: 38776 | OCT4 promoter from phOct4-EGFP (Addgene) was subcloned into pCLuc-Basic2 |
| Antibody | CD326 (EpCAM)-FITC | Miltenyi Biotec | Catalog number: 130-113-263; RRID:AB_2726064 | Application: flow cytometry |
| Antibody | BD Pharmingen APC Mouse Anti-Human CD44 | BD Biosciences | Catalog number: 559942; RRID:AB_398683 | Application: flow cytometry |
| Antibody | BD Pharmingen PE Mouse Anti-Human CD24 | BD Biosciences | Catalog number: 555428; RRID:AB_395822 | Application: flow cytometry |
| Antibody | H-2Kd/H-2Dd clone 34-1-2S | SouthernBiotech | Catalog number: 1911–08; RRID:AB_1085008 | Application: flow cytometry |
| Antibody | Anti-HNF1 antibody [GT4110] | Abcam | Catalog number: ab184194; RRID:AB_2538735 | Application: IHC, Western blot |
| Antibody | HNF-1 alpha Antibody (C-19) | Santa Cruz Biotechnology | Catalog number: sc-6547; RRID:AB_648295 | ChIP |
| Antibody | Normal Rabbit IgG | Cell Signaling Technology | Catalog number: 2729S; RRID:AB_1031062 | ChIP |

*Continued on next page*

*Continued*

| Reagent type (species) or resource | Designation | Source or reference | Identifiers | Additional information |
|---|---|---|---|---|
| Antibody | HNF1α (D7Z2Q) | Cell Signaling Technology | Catalog number: 89670S; RRID:AB_2728751 | Application: Western blot |
| Antibody | β-Actin (clone AC-74) | Sigma Aldrich | Catalog number: A2228-200UL; RRID:AB_476697 | Application: Western blot |
| Antibody | CDH17 antibody | Proteintech | Catalog number: 50-608-369; RRID:AB_2728752 | Application: Western blot |
| Antibody | DPP4/CD26 (D6D8K) | Cell Signaling Technology | Catalog number: 67138S; RRID:AB_2728750 | Application: Western blot |
| Antibody | CD44 (156–3 C11) | Cell Signaling Technology | Catalog number: 3570S; RRID:AB_10693293 | Application: Western blot |
| Antibody | EpCAM (D1B3) | Cell Signaling Technology | Catalog number: 2626S; RRID:AB_2728749 | Application: Western blot |
| Antibody | Cleaved Caspase-3 (Asp175) (5A1E) | Cell Signaling Technology | Catalog number: 9664S; RRID:AB_2070042 | Application: Western blot |
| Antibody | Cleaved Caspase-6 (Asp162) | Cell Signaling Technology | Catalog number: 9761S; RRID:AB_2290879 | Application: Western blot |
| Antibody | Cleaved Caspase-7 (Asp198) (D6H1) | Cell Signaling Technology | Catalog number: 8438S; RRID:AB_11178377 | Application: Western blot |
| Antibody | Cleaved Caspase-9 (Asp330) (D2D4) | Cell Signaling Technology | Catalog number: 7237S; RRID:AB_10895832 | Application: Western blot |
| Antibody | Cleaved Caspase-9 (Asp315) Antibody | Cell Signaling Technology | Catalog number: 9505S; RRID:AB_2290727 | Application: Western blot |
| Antibody | GFP (D5.1) XP | Cell Signaling Technology | Catalog number: 2956S; RRID:AB_1196615 | Application: Western blot |
| Antibody | Ras (G12D Mutant Specific) (D8H7) | Cell Signaling Technology | Catalog number: 14429S; RRID:AB_2728748 | Application: Western blot |
| Antibody | Phospho-p44/42 MAPK (Erk1/2) (Thr202/Tyr204) (D13.14.4E) XP | Cell Signaling Technology | Catalog number: 4370S; RRID:AB_2315112 | Application: Western blot |
| Antibody | Phospho-Akt (Ser473) (D9E) XP | Cell Signaling Technology | Catalog number: 4060S; RRID:AB_2315049 | Application: Western blot |
| Antibody | Oct-4A (C52G3) | Cell Signaling Technology | Catalog number: 2890S; RRID:AB_2167725 | Application: Western blot |
| Antibody | Anti-KRAS + HRAS + NRAS antibody | Abcam | Catalog number: ab55391; RRID:AB_941040 | Application: Western blot |
| Antibody | Anti-β-Galactosidase | Promega | Catalog number: Z3781; RRID:AB_430877 | Application: Western blot |

*Continued on next page*

*Continued*

| Reagent type (species) or resource | Designation | Source or reference | Identifiers | Additional information |
|---|---|---|---|---|
| Antibody | IRDye 800CW Goat anti-Mouse IgG | Licor | Catalog number: 926–32210; RRID:AB_621842 | Application: Western blot |
| Antibody | IRDye 800CW Goat anti-Rabbit | Licor | Catalog number: 926–32211; RRID:AB_621843 | Application: Western blot |
| Antibody | IRDye 680LT goat anti-mouse | Licor | Catalog number: 926–68020; RRID:AB_10706161 | Application: Western blot |
| Antibody | IRDye 680LT Goat anti-Rabbit IgG | Licor | Catalog number: 926–68021; RRID:AB_10706309 | Application: Western blot |
| Recombinant DNA reagent | pLentipuro3/TO/V5-DEST | Andrew E. Aplin, Thomas Jefferson University | | |
| Recombinant DNA reagent | pLentineo3/TO/V5-DEST | Andrew E. Aplin, Thomas Jefferson University | | |
| Recombinant DNA reagent | pLentihygro3/TO/V5-DEST | Andrew E. Aplin, Thomas Jefferson University | | |
| Recombinant DNA reagent | pLenti0.3/EF/V5-DEST | This paper | | Human EF1-alpha promoter was substituted for the CMV promoter of pLenti6.3/UbC/V5-DEST and the SV40 promoter/Bsd cassette was removed |
| Recombinant DNA reagent | pLenti6.3/UbC/V5-DEST | Andrew E. Aplin, Thomas Jefferson University | | |
| Recombinant DNA reagent | pLenti6.3/UbC empty vector | This paper | | EcoRV digest/re-ligation to remove Gateway element |
| Recombinant DNA reagent | pLentipuro3/Block-iT-DEST | Andrew E. Aplin, Thomas Jefferson University | | |
| Recombinant DNA reagent | pLenti0.3/EF/GW/IVS-Kozak-TetR-P2A-Bsd | This paper | | LR recombination of IVS-TetR-P2A-Bsd cassette into pLenti0.3/EF/V5-DEST |
| Recombinant DNA reagent | pLenti0.3/EF/GW/PatGFP-Luc2 | This paper | | LR recombination of PatGFP-Luc2 cassette into pLenti0.3/EF/V5-DEST |
| Recombinant DNA reagent | pLP1 | Andrew E. Aplin, Thomas Jefferson University | | Lentivirus packaging plasmid originally from Invitrogen |
| Recombinant DNA reagent | pLP2 | Andrew E. Aplin, Thomas Jefferson University | | Lentivirus packaging plasmid originally from Invitrogen |
| Recombinant DNA reagent | pLP/VSVG | Andrew E. Aplin, Thomas Jefferson University | | Lentivirus packaging plasmid originally from Invitrogen |
| Sequence-based reagent | Non-targeting control siRNA | Dharmacon | Catalog number: D-001810-01-20 | |
| Sequence-based reagent | HNF1A siRNA #1 | Dharmacon | Catalog number: D-008215-01-0002 | Sequence: GGAGGAACCGTTTCAAGTG |
| Sequence-based reagent | HNF1A siRNA #2 | Dharmacon | Catalog number: D-008215-02-0002 | Sequence: GCAAAGAGGCACTGATCCA |

*Continued on next page*

*Continued*

| Reagent type (species) or resource | Designation | Source or reference | Identifiers | Additional information |
|---|---|---|---|---|
| Sequence-based reagent | POU5F1/OCT4 siRNA #5 | Dharmacon | Catalog number: D-019591-05-0002 | Sequence: CATCAAAGCTCTGCAGAAA |
| Sequence-based reagent | POU5F1/OCT4 siRNA #6 | Dharmacon | Catalog number: D-019591-06-0002 | Sequence: GATATACACAGGCCGATGT |
| Sequence-based reagent | POU5F1/OCT4 siRNA #9 | Dharmacon | Catalog number: D-019591-09-0002 | Sequence: GCGATCAAGCAGCGACTAT |
| Sequence-based reagent | POU5F1/OCT4 siRNA #10 | Dharmacon | Catalog number: D-019591-10-0002 | Sequence: TCCCATGCATTCAAACTGA |
| Peptide, recombinant protein | Recombinant human EGF | Invitrogen | Catalog number: PHG0311L | |
| Peptide, recombinant protein | FGF-basic Recombinant Human | Invitrogen | Catalog number: PHG0264 | |
| Peptide, recombinant protein | Leukemia Inhibitory Factor human | Sigma Aldrich | Catalog number: L5283 | |
| Peptide, recombinant protein | Bone Morphogenetic Protein four human | Peprotech | Catalog number: 120–05 | |
| Commercial assay or kit | SimpleChIP Enzymatic Chromatin IP Kit (Magnetic Beads) | Cell Signaling Technology | Catalog number: 9003 | |
| Commercial assay or kit | BioLux *Gaussia* Luciferase Assay Kit | New England Biolabs | Catalog number: E3300S | |
| Commercial assay or kit | BioLux *Cypridina* Luciferase Assay Kit | New England Biolabs | Catalog number: E3309S | |
| Commercial assay or kit | RNeasy Plus Mini Kit coupled with RNase-free DNase set | Qiagen | Catalog number: 74136 and 79254 | |
| Commercial assay or kit | High Capacity RNA-to-cDNA Master Mix | Applied Biosystem | Catalog number: 4387406 | |
| Commercial assay or kit | Power SYBR Green PCR Master Mix | Applied Biosystem | Catalog number: 4367659 | |
| Chemical compound, drug | APC-Cy7 Streptavidin | BD Biosciences | Catalog number: 554063 | |
| Chemical compound, drug | DAPI (4',6-Diamidino-2-Phenylindole, Dilactate) | Invitrogen | Catalog number: 3571 | |
| Chemical compound, drug | APC Annexin V | BD Biosciences | Catalog number: 550474 | |
| Chemical compound, drug | Annexin V Binding Buffer, 10x concentrate | BD Biosciences | Catalog number: 556454 | |
| Chemical compound, drug | RNase A | Invitrogen | Catalog number: 12091021 | |
| Chemical compound, drug | Lipofectamine 2000 Reagent | Invitrogen | Catalog number: 11668019 | |
| Chemical compound, drug | Lipofectamine RNAiMAX Reagent | Invitrogen | Catalog number: 13778150 | |
| Chemical compound, drug | Propidium iodide | Invitrogen | Catalog number: P1304MP | |
| Chemical compound, drug | Gentamicin | Invitrogen | Catalog number: 15710072 | |

*Continued on next page*

*Continued*

| Reagent type (species) or resource | Designation | Source or reference | Identifiers | Additional information |
|---|---|---|---|---|
| Chemical compound, drug | Antibiotic-Antimycotic (100X) | Invitrogen | Catalog number: 15240062 | |
| Chemical compound, drug | N-2 Supplement (100X) | Invitrogen | Catalog number: 17502–048 | |
| Chemical compound, drug | B-27 Serum-Free Supplement (50X) | Invitrogen | Catalog number: 17504–044 | |
| Chemical compound, drug | Doxycycline | Sigma Aldrich | D9891-100G | |
| Software, algorithm | GraphPad Prism 6 | GraphPad Software; http://www.graphpad.com | RRID:SCR_002798 | |
| Software, algorithm | oPOSSUM 3.0 | http://opossum.cisreg.ca/oPOSSUM3/; PMID: 22973536 | RRID:SCR_010884 | |
| Software, algorithm | Bowtie v1.1.1 | PMID: 19261174 | RRID:SCR_005476 | |
| Software, algorithm | Bowtie v0.12.8 | PMID: 19261174 | RRID:SCR_005476 | |
| Software, algorithm | MACS v1.4.2 | PMID: 18798982 | RRID:SCR_013291 | |
| Software, algorithm | TopHat v1.4.1 | PMID: 19289445 | RRID:SCR_013035 | |
| Software, algorithm | DESeq v1.24.0 | PMID: 20979621 | RRID:SCR_000154 | |
| Software, algorithm | bedtools v.2.26.0 | PMID: 20110278 | RRID:SCR_006646 | |
| Software, algorithm | survival v2.40–1 | DOI: 10.1007/978-1-4757-3294-8 | | |

## Tumor growth assays

Eight- to 10-week-old, evenly sex-mixed NOD/SCID mice were used for all experiments. Orthotopic implantation of PDA cells to the pancreas has previously been described (*Abel et al., 2014*). Briefly, mice were anesthetized with an intraperitoneal injection of 100 mg/kg ketamine/5 mg/kg xylazine, and a small left subcostal incision was performed. 10,000 PatGFP-Luc2-labeled tumor cells in a volume of 50 µl (1:1 vol of cell suspension in growth media and Matrigel) were injected into the tail of the pancreas using a 30-gauge needle. Weekly bioluminescent imaging of implanted orthotopic tumors in mice was performed using a Xenogen IVIS 200 Imaging System (Xenogen Biosciences, Cranbury, NJ). For subcutaneous implantation of tumor cells, 10,000 tumor cells in a volume of 50 µl (1:1 vol of cell suspension in growth media and Matrigel) was injected subcutaneously into both the left and right midflank regions of mice. Tumor growth was monitored weekly by digital caliper and tumor volumes calculated by the (length x width$^2$)/2 method. All mice were sacrificed once any tumors reached 20 mm$^3$ in volume.

## Immunofluorescence and immunohistochemistry

Formalin-fixed, paraffin-embedded tumor samples were sectioned and processed for immunofluorescent staining by the University of Michigan ULAM Pathology Cores for Animal Research. Immunohistochemistry was performed using a Ventana BenchMark Ultra autostainer. HNF1A antibody (GT4110) was used for immunohistochemistry at a 1:100 dilution. A PDA/normal pancreas tissue microarray was generated by the University of Michigan Department of Pathology.

## Microscopy

All microscopies were performed on an Olympus IX83 motorized inverted microscope with cellSens Dimension software (Olympus Corporation, Waltham, MA).

## Lentiviral constructs

Lentiviral destination vectors were generously provided by Dr. Andrew Aplin (Thomas Jefferson University). For construction of HNF1A, KRAS$^{G12D}$, GFP and LacZ cDNA lentiviruses, pLentipuro3/TO/V5-DEST, pLentineo3/TO/V5-DEST, pLentihygro3/TO/V5-DEST were used. For OCT4A, pLenti6.3/UbC/V5-DEST was used. An EcoRV digested/re-ligated pLenti6.3/UbC/V5-DEST (removing the Gateway cloning element) was used as an empty vector control. For construction of shRNA lentiviruses, pLentipuro3/BLOCK-iT-DEST was used. Human HNF1A and KRAS$^{G12D}$ were cloned from primary PDA cDNA into pENTR/D-TOPO (Invitrogen). Human OCT4A was cloned from pCR4-TOPO clone BC117435 (Transomic Technologies) into pENTR/D-TOPO. LacZ and PatGFP (a variant of EGFP containing the following mutations: S31R, Y40N, S73A, F100S, N106T, Y146F, N150K, M154T, V164A, I168T, I172V, A207V) were also cloned into pENTR/D-TOPO as control proteins. For labeling cells with firefly luciferase, PatGFP was fused to the N-terminus of firefly luciferase Luc2 (subcloned from pGL4.10) and cloned into pENTR/D-TOPO using Gibson Assembly (New England Biolabs). PatGFP-Luc2 was recombined into pLenti0.3/EF/V5-DEST, a modified version of pLenti6.3/UbC/V5-DEST with the human EF-1α promoter instead of the human UbC promoter and no downstream promoter/selective marker cassette, to generate pLenti0.3/EF/GW/PatGFP-Luc2. To generate doxycycline-inducible cell lines, a cassette containing the IVS-TetR region from pLenti6/TR (Invitrogen) was subcloned into pLenti0.3/EF/V5-DEST, along with a C-terminal P2A peptide-blasticidin resistance gene (Bsd) reading frame to generate pLenti0.3/EF/GW/IVS-Kozak-TetR-P2A-Bsd. The resultant lentiviruses were used to transduce NY8, NY15, NY53, and HPDE to generate doxycycline-inducible 'TR' lines. To generate the HNF1A-responsive reporter, the multiple cloning site and minimal promoter from pTA-Luc (Takara, Mountain View, CA) was subcloned upstream of PatGFP. Eight tandem repeats of the HNF1A-binding site with spacer nucleotides (CTTGGTTAATGATTAACCAGA) was cloned between the MluI and BglII sites of the multiple cloning site. LacZ2.1 (CACCAAATCGCTGATTTGTGTAGTCGTTCAAGAGACGACTACACAAATCAGCGA), HNF1A shRNA#1 (CACCGCTAGTGGAGGAGTGCAATTTCAAGAGAATTGCACTCCTCCACTAGC), and HNF1A shRNA#2 (CACCGTCCCTTAGTGACAGTGTCTATTCAAGAGATAGACACTGTCACTAAGGGAC) were cloned into pENTR/H1/TO (Invitrogen). cDNA and shRNA constructs were recombined into their respective lentiviral plasmids using LR Clonase II (Invitrogen). The resulting constructs were packaged in 293FT cells as previously described.

## siRNA sequences

Non-targeting control (Cat#D-001810–01)
 HNF1A-targeting siRNA#1 (GGAGGAACCGTTTCAAGTG)
 HNF1A-targeting siRNA#2 (GCAAAGAGGCACTGATCCA)
 POU5F1/OCT4-targeting siRNA#5 (CATCAAAGCTCTGCAGAAA)
 POU5F1/OCT4-targeting siRNA#6 (GATATACACAGGCCGATGT)
 POU5F1/OCT4-targeting siRNA#9 (GCGATCAAGCAGCGACTAT)
 POU5F1/OCT4-targeting siRNA#10 (TCCCATGCATTCAAACTGA)

## Cell lines

HPDE cells were a generous gift from Dr. Craig Logsdon (MD Anderson). HPNE, Capan-2, HPAF-II, BxPC-3, AsPC-1, Panc-1, and MiaPaCa-2 cells were purchased from ATCC (Manassas, VA). For all low-passage human primary PDA cells, primary PDA xenograft tumors were cut into small pieces with scissors and then minced completely using sterile scalpel blades. Single cells were obtained described previously (*Li et al., 2007*). The cells used in this article are passaged less than 10 times in vitro. All cells were authenticated by STR profiling (University of Michigan DNA Sequencing Core). Cells were routinely tested for mycoplasma contamination using the MycoScope PCR Detection kit (Genlantis, San Diego, CA) and only mycoplasma-free cells were used for experimentation. ATCC and primary PDA cells were cultured in RPMI-1640 with GlutaMAX-I supplemented with 10% FBS (Gibco), 1% antibiotic-antimycotic (Gibco), and 100 μg/ml gentamicin (Gibco). HPDE cells were maintained in keratinocyte SFM supplemented (Invitrogen) with included EGF and bovine pituitary extract as well as 1% antibiotic-antimycotic and 100 μg/ml gentamicin.

## Soft agar assays

Low-melting agarose (Invitrogen) was dissolved in serum-free RPMI-1640 with GlutaMAX-I to a final concentration of 2% at 60°C and cooled to 42°C. 200 μL per well 2% agarose was evenly spread at the bottom of a 24-well dish, followed by 250 μL of 0.6% agarose (diluted with complete keratinocyte SFM and supplemented with FBS to 2.5%), a 250 μL of 0.4% agarose/cell suspension, and a 250 μL of acellular 0.4% agarose. Each layer was allowed to solidify a 4°C for 10 min and then heated to 37°C prior to adding the next layer. 500 μl of complete keratinocyte SFM and supplemented with 2.5% FBS was added atop each gel and replenished every 3 days.

## Flow cytometry

Flow cytometry was performed as described previously (*Li et al., 2007*). Cells were dissociated with 2.5% trypsin/EDTA solution, counted and transferred to 5 mL tubes, washed with HBSS supplemented with FBS twice and resuspended in HBSS/2% FBS at a concentration of 1 million cells/100 μL. Primary antibodies were diluted 1:40 in cell suspensions and incubated for 30 min on ice with occasional vortexing. Cells were washed twice with HBSS/2% FBS and incubated for 20 min on ice with APC-Cy7 Streptavidin diluted 1:200. Cells were washed twice with HBSS/2% FBS and resuspended in HBSS/2%FBS containing 3 μM 4',6-diamidino-2-phenylindole (DAPI) (Invitrogen, Carlsbad, CA). Flow cytometry and sorting was done using a FACSAria (BD Biosciences, Franklin Lakes, NJ). Side scatter and forward scatter profiles were used to eliminate cell doublets, APC-Cy7 was used to exclude mouse cells. For PatGFP-Luc2 labeling, GFP+/DAPI- cells were isolated by sorting and expanded for one passage prior to implantation. For analysis of apoptosis, APC-conjugated Annexin V and Annexin V binding buffer (BD Biosciences) was used following manufacturer's recommendations with 3 μM DAPI added immediately before analysis to stain permeable cells/necrotic debris.

## Propidium iodide staining

Cells were trypsinized, washed in PBS and fixed in 70% ethanol for 4 hr. Cells were then permeabilized with PBS containing 0.1% Triton X100 and 200 μg/ml RNase A for 2 hr at 37°C and stained with 167 μg/ml propidium iodide for 30 min. DNA content was measured by flow cytometry on a Cyto-FLEX flow cytometer (Beckman Coulter) and analyzed Summit v6.2 software (Beckman Coulter).

## Microarray analysis

Flow sorted NY8 and NY15 P1, P2, and P3 cells were immediately used for RNA isolation using the RNeasy Plus Mini Kit coupled with RNase-free DNase set (Qiagen). Microarrays and analyses were performed by the University of Michigan DNA Sequencing Core. RNA labeling and hybridization was conducted using the Human Genome U133 Plus 2.0 microarray (Affymetrix, Santa Clara, CA). Probe signals were normalized and corrected according to background signal. Adjusted signal strength was used to generate quantitative raw values, which were log-transformed for all subsequent analyses.

## Transcription-factor-binding site analysis

For both the PCSC-enriched genes (related to *Figure 1*) and the HNF1A target genes (related to *Figure 7*), oPOSSUM 3.0 (http://opossum.cisreg.ca/oPOSSUM3/) (*Kwon et al., 2012*) was used to detect over-represented conserved transcription factor binding sites. The program was run using the following options: conservation cutoff of 0.4, matrix score threshold of 85%, and search region of 5 kbp, upstream and downstream of the transcription start site. The query was entered against a background of 24,752 genes in the oPOSSUM database.

## Quantitative reverse transcription-PCR (qRT-PCR)

Total RNA was extracted using RNeasy Plus Mini Kit coupled with RNase-free DNase set (Qiagen) and reverse transcribed with High Capacity RNA-to-cDNA Master Mix

(Applied Biosystem). The resulting cDNAs were used for PCR using Power SYBR Green PCR Master Mix (Applied Biosystem) in triplicates. qPCR and data collection were performed on a ViiA7 Real-Time PCR system (Invitrogen). Conditions used for qPCR were 95°C hold for 10 min, 40 cycles of 95°C for 10 s, 60°C for 15 s, and 72°C for 20 s. All quantitations were normalized to an endogenous control *ACTB*. The relative quantitation value for each target gene compared to the calibrator for

that target is expressed as 2-(Ct-Cc) (Ct and Cc are the mean threshold cycle differences after normalizing to *ACTB*).

## Tumorsphere cultures

Single cells were suspended in tumorsphere culture media containing 1% N2 supplement, 2% B27 supplement, 1% antibiotic-antimycotic, 20 ng/mL epidermal growth factor (Gibco, Carlsbad, CA), 20 ng/mL human bFGF-2 (Invitrogen), 10 ng/mL BMP4 (Sigma-Aldrich, St. Louis, MO), 10 ng/mL LIF (Sigma-Aldrich) and plated in six-well Ultra-Low Attachment Plates (Corning, Corning, NY).

siRNA transfection siRNA were purchased from Dharmacon (Lafayette, CO) and were transfected at 25 nM using Lipofectamine RNAiMAX Reagent (Invitrogen). siRNA sequences can be found in the Supplementary Material and methods.

## Western blotting

All lysates were boiled in 1x Laemmli sample buffer with β-mercaptoethanol for 5 min followed by electrophoresis on 4–20% Mini-PROTEAN TGX precast Tris-Glycine-SDS gels (Bio-Rad, Hercules, CA). Proteins were transferred to low-fluorescent PVDF (Bio-Rad) and incubated overnight in primary antibody at 1:1000 dilution. Blots were incubated in IRDye-conjugated secondary antibodies at room temperature for 1 hr and imaged/quantitated by an Odyssey CLx imaging system (Li-Cor, Lincoln, NE). For western blotting, HNF1A (clone GT4110) and KRAS (ab55391) from Abcam (Cambridge, MA), β-Actin (clone AC-74) from Sigma-Aldrich, Cadherin-17 (CDH17) from Proteintech (Rosemont, IL), β-Galactosidase from Promega (Madison, WI) and RAS$^{G12D}$, CD44, EPCAM, DPP4, Cleaved Caspase-3 (D175), Cleaved Caspase-6 (D162), Cleaved Caspase-7 (D198), Cleaved Caspase-9 (D315), Cleaved Caspase-9 (D330), phospho-ERK1/2, phospho-AKT S473, OCT4A and GFP from Cell Signaling Technology (Danvers, MA). For flow cytometry, mouse anti-human EPCAM (CD326) clone HEA-125 was purchased from Miltenyi Biotec (San Diego, CA). Mouse anti-human CD44 clone G44-26, CD24 clone ML5 and APC-Cy7 Streptavidin were purchased from BD Biosciences (San Jose, CA). Biotinylated mouse anti-mouse H-2Kd/H-2Dd clone 34-1-2S was purchased from SouthernBiotech (Birmingham, AL).

## Reporter assays

For the Cypridina luciferase construct containing the full-length canonical OCT4 promoter, a 3.9-kbp insert was excised from phOct4-EGFP (*Gerrard et al., 2005*) by XhoI and BamHI digestion, followed by ligation into pCLuc-Basic2 (New England Biolabs). phOct4-EGFP was a gift from Wei Cui (Addgene plasmid # 38776). For the POU5F1/OCT4 LTR construct, a 1.7-kbp insert was amplified from NY5 genomic DNA with the following primers: 5'-ATCTTGGAATTCTGGGCACTCAGTTTATTG TTAGG-3' and 5'-GGTGGCGGATCCTGTGTTAATCCTCCTCGGGG-3'. The insert was digested with EcoRI and BamHI and cloned into pCLuc-Basic2. Cypridina luciferase constructs were co-transfected with Lipofectamine 2000 (Invitrogen) into 293FT cells with either LacZ or HNF1A lentiviral expression plasmids and the internal control plasmid pTK-GDLuc, a variant of pTK-GLuc (New England Biolabs) in which the Gaussia luciferase coding region was replaced with the coding region for Gaussia-Dura (Millipore) in order to provide a more stable luciferase signal. Cypridina and Gaussia-Dura luciferase activities were measured in conditioned media 48 hr post-transfection with the BioLux Cypridina Luciferase and BioLux Gaussia Luciferase Assay Kits (New England Biolabs), respectively.

## Chromatin immunoprecipitation sequencing (ChIP-seq)

A confluent 15 cm culture plate of cells was used per immunoprecipitation. Cells were fixed with 1% formaldehyde for 10 min. Nuclei were collected and chromatin sheared to 1–10 nucleosomes using the SimpleChIP Plus Enzymatic Chromatin IP kit and protocol (Cell Signaling). HNF1A was immunoprecipitated with goat polyclonal antibody C-19 (Santa Cruz). Libraries from HNF1A-immunoprecipitated chromatin and input chromatin was prepared by the University of Michigan Sequencing Core and sequenced on the Illumina HiSeq 4000.

## Chromatin Immunoprecipitation-PCR

Chromatin was prepared as indicated for ChIP-seq and immunoprecitated with either normal goat IgG (R and D Systems) or anti-HNF1A (C-19, Santa Cruz Biotechnology) overnight using the

SimpleChIP Plus Enzymatic Chromatin IP kit and protocol. Quantitative PCR was performed using immunoprecipitated DNA and 2% chromatin input DNA as described earlier for qRT-PCR using modified thermocycling conditions: 95°C hold for 10 min, 45 cycles of 95°C for 15 s and 60°C for 60 s. Percent Input for immunoprecipitated DNA was calculated using the formula 2% x $2^{(\text{Ct 2\% Input Sample - Ct IP Sample})}$. Primers for POU5F1/OCT4 regulatory regions were as follows: half-site #1 (HS1) (5'-GTGAAATCTTTAGTGTTGTGAG-3' and 5'-CCAAGAAATGTAGCAGGACGAGCCCC-3'), half-site #2 (HS2) (5'-AACCTTTTACATGAGCAGGTTTG-3' and 5'-AATGGTGGAAAGAATTACATGG-3'), half-site #3 (HS3) (5'-GGGCACTCAGTTTATTGTTAGG-3' and 5'-TTTCCTGTCACAGGGGTTTAGTG-3'), and distal enhancer (DE) (5'-GAGAGGCCGTCTTCTTGGCAGAC-3' and 5'-GTTCACTTCTCGGCCTTTAACTGCCC-3'). MYOD (primers 5'-AGACTGCCAGCACTTTGCTATC-3' and 5'-ATAGAAGTCGTCCGTTGTGGC-3') was used as a non-HNF1A target gene control.

## Bromouridine labeling and sequencing (Bru-seq)

Nascent RNA labeling and sequencing (Bru-seq) was performed as previously described (*Paulsen et al., 2013*). For each shRNA (LacZ2.1, HNF1A shRNA#1, and #2), two replicates were performed in each cell line (NY8 and NY15). Cells were incubated in media containing 2 mM bromouridine (Bru) (Aldrich) for 30 min at 37°C. Total RNA was isolated after lysis in Trizol and Bru-RNA was isolated using anti-BrdU antibodies conjugated to magnetic beads. Strand-specific libraries were made using the Illumina TruSeq kit and sequenced on the Illumina HiSeq 4000 platform at the University of Michigan Sequencing Core (Ann Arbor, MI). Genes were recognized as differentially expressed in both cell lines if the fold change after knockdown was greater than 1.5 (and FDR < 0.1 in NY15) and the mean RPKM for a given comparison was greater than 0.25 in either HNF1A shRNA#1 or shRNA#2 per cell line.

## ChIP-seq analysis

The HNF1A ChIP-seq experiment consisted of 2 replicates each of input and ChIP libraries from both NY8 and NY15 cells (eight libraries altogether). 52-base, single end reads were aligned to the human reference genome (hg19) using Bowtie v1.1.1 (with options: -n 3 k 1 m 1). Peaks were called using MACS v1.4.2 using the default options and input samples as controls. MACS peaks overlapping ENCODE blacklist regions (https://www.encodeproject.org/annotations/ENCSR636HFF) were removed. Peak counts were 5057 (NY15 rep1), 8616 *NY15 rep2), 64603(NY8 rep1), and 13169 (NY8 rep2). Each peak was assigned to the closest expressed gene's transcription start site (TSS). Then, for each TSS, the distance to the nearest peak was measured. If the nearest associated peak was within ±5 kb of the TSS, it was considered proximal. In the absence of a proximal peak, the nearest associated peak within ±100 kb of the TSS was considered distal. A gene was recognized as having a proximal or distal peak if at least one replicate in both cell lines identified a proximal or distal peak. If a gene was found to have both proximal and distal peaks (usually due to differences between replicates), the gene was identified as distal if it had distal peaks in both replicates of both cell lines, otherwise it was identified as neither. Manipulation of genomic regions was performed using bedtools2 (v2.26.0).

## Bru-seq analysis

The HNF1A knockdown experiment used for Bru-seq consisted of a control shRNA and two different HNF1A-targeting shRNAs for each of NY8 and NY15 cells, and 2 replicates of each (12 samples altogether). 52-base, stranded, single end reads were aligned first to ribosomal DNA (U13369.1) using Bowtie v0.12.8 and the remaining reads aligned to the human reference genome (hg19) using TopHat v1.4.1. Differential gene expression analysis was performed using DESeq v1.24.0 (R v3.3.1). Gene annotation and counting was performed as previously described (*Paulsen et al., 2014*). Differentially expressed genes were selected based on the following criteria: mean RPKM >0.25 across samples, minimum gene length 300, absolute value of log2 fold-change >0.58 (1.5 fold-change), adjusted p value<0.1, and these requirements met for at least one HNF1A shRNA in both cell lines.

## Data access

All ChIP-seq and Bru-seq data from this study are available at the NCBI Gene Expression Omnibus (GEO; accession # GSE108151).

## Enhancer-related analysis

Enhancer regions used in this study were taken from the ENCODE Combined Segmentation annotation (http://hgdownload.soe.ucsc.edu/goldenPath/hg19/encodeDCC/wgEncodeAwgSegmentation/) (*Hoffman et al., 2013*; *Ernst and Kellis, 2012*; *Hoffman et al., 2012*). Regions labeled 'E' (strong enhancers) were extracted from all six cell lines used in the Combined Segmentation analysis, then merged to create a set of general putative enhancer regions. The enhancer regions were then queried against peak coordinated from each list of ChIP-seq peaks (see *Supplemental file 2*). All genomic region manipulations were performed using bedtools2 (v.2.26.0).

## Survival analysis

Gene expression and patient survival data for pancreatic adenocarcinoma were obtained through the Broad Institute TCGA Genome Data Analysis Center (PAAD cohort; 2016; Firehose stddata__2016_01_28 run; Broad Institute of MIT and Harvard; doi:10.7908/C11G0KM9). Clinical metadata were obtained from both the Merge Clinical Level one and Clinical Pick Tier 1 Level four data sets. Gene expression values were obtained from the Level 3 RSEM genes (normalized) data set and $\log_{10}$-transformed prior to analysis (a constant of 1 added to preserve zeros). Samples identified as primary solid tumors and of non-neuroendocrine origin were used. Specifically, samples with the following values in the 'patient.histological_type_other' field were rejected: '82463 neuroendocrine carcinoma nos', 'moderately differentiated ductal adenocarcinoma 60% + neuroendocrine 40%', 'neuroendocrine', 'neuroendocrine carcinoma', and 'neuroendocrine carcinoma nos'. The background set of genes were defined as those with Bru-seq RPKM greater than 0.5 in at least one replicate of both NY8 and NY15 cells and which mapped to either gene symbol or entrez gene ID in the TCGA expression data. Cox proportional hazards survival models were created using the R package survival (v2.40–1). For permutation testing against a particular set of HNF1A-related genes, random sets of genes of the same size were selected from the background set and the percent of genes significantly associated with reduced or increased survival (using FDR thresholds of 0.1 and 0.25) were calculated. In order for the estimated error of the estimated p value to be less than 10% (at significant level $\alpha = 0.05$), we set the number of permutations (N) to 10,000.

## Other statistical analysis

The following methods are specific to analysis of the data represented in *Figures 1–6* and *Figure 1—figure supplement 2*, *Figure 2—figure supplement 1*, *Figure 3—figure supplement 1*, *Figure 4—figure supplement 1*, *Figure 6—figure supplement 1*, and *Figure 6—figure supplement 2*. Data are expressed as the mean ±SEM. Statistically significant differences between two groups was determined by the two-sided Student t-test for continuous data, while ANOVA was used for comparisons among multiple groups. Significance was defined as p<0.05. GraphPad Prism six was used for these analyses.

## Study approval

All animal protocols were approved by University Committee for the Use and Care of Animals (UCUCA) at University of Michigan. The animal welfare assurance number for this study is A3114-01. Patient samples were collected under a protocol approved by the IRB at the The University of Michigan. All patients gave informed consent. The human assurance number for this study is FWA00004969.

## Acknowledgements

We thank the University of Michigan Flow Cytometry Core facility for assistance with performing FACS analysis and sorting, the University of Michigan DNA Sequencing Core facility for assistance ChIP-seq and microarray setup and analysis, Armand Bankhead for bioinformatic consultation, and Michelle Paulsen for preparing samples for Bru-seq. The work was supported by the Pancreatic Cancer Action Network-AACR Pathway to Leadership Grant (16-70-25-ABEL) and the American Cancer Society Postdoctoral Fellowship (127662-PF-15-033-01-DDC) (to EVA), University of Michigan Comprehensive Cancer Center Core Grant (P30 CA046592) (to BM), and the Gershenson Pancreatic Cancer Fund (DMS) and SKY Foundation (HC and DMS).

## Additional information

### Funding

| Funder | Grant reference number | Author |
|---|---|---|
| American Cancer Society | 127662-PF-15-033-01-DDC | Ethan V Abel |
| Pancreatic Cancer Action Network | 16-70-25-ABEL | Ethan V Abel |
| University of Michigan Comprehensive Cancer Center | Core Grant P30 CA046592 | Brian Magnuson |
| SKY Foundation | | Howard C Crawford<br>Diane M Simeone |
| Gershenson Pancreatic Cancer Fund | | Diane M Simeone |

The funders had no role in study design, data collection and interpretation, or the decision to submit the work for publication.

### Author contributions

Ethan V Abel, Conceptualization, Resources, Formal analysis, Funding acquisition, Validation, Investigation, Visualization, Methodology, Writing—original draft, Project administration, Writing—review and editing; Masashi Goto, Conceptualization, Validation, Investigation, Methodology; Brian Magnuson, Resources, Data curation, Software, Formal analysis, Validation, Visualization, Methodology, Writing—original draft, Writing—review and editing; Saji Abraham, Nikita Ramanathan, Emily Hotaling, Anthony A Alaniz, Michele L Dziubinski, Sumithra Urs, Lidong Wang, Investigation; Chandan Kumar-Sinha, Resources, Formal analysis, Validation; Jiaqi Shi, Resources, Investigation; Meghna Waghray, Supervision; Mats Ljungman, Resources, Methodology; Howard C Crawford, Supervision, Funding acquisition, Investigation, Project administration, Writing—review and editing; Diane M Simeone, Conceptualization, Resources, Supervision, Funding acquisition, Writing—original draft, Project administration, Writing—review and editing

### Author ORCIDs

Ethan V Abel (iD) http://orcid.org/0000-0003-2922-617X
Brian Magnuson (iD) http://orcid.org/0000-0002-5301-3302
Diane M Simeone (iD) https://orcid.org/0000-0001-5142-3087

### Ethics

Human subjects: Patient samples were collected under a protocol approved by the IRB at the The University of Michigan. All patients gave informed consent. The human assurance number for this study is FWA00004969.
Animal experimentation: All animal protocols were approved by University Committee for the Use and Care of Animals (UCUCA) at The University of Michigan. The animal welfare assurance number for this study is A3114-01. Every effort was made throughout this study to minimize stress to and suffering of animal subjects.

### Decision letter and Author response

Decision letter https://doi.org/10.7554/eLife.33947.041
Author response https://doi.org/10.7554/eLife.33947.042

## Additional files

### Supplementary files

• Supplementary file 1. Cancer stem cell frequencies in PDA subpopulations. Limiting dilution assay was performed with sorted NY15 cells injected subcutaneously in NOD/SCID mice. The resultant

numbers of tumors/injection is tabulated with estimated cancer stem cell frequencies calculated by extreme limiting dilution analysis (ELDA).

DOI: https://doi.org/10.7554/eLife.33947.030

• Supplementary file 2. Data for generating PDA subpopulation heatmap and HNF1A target gene data (Excel spreadsheet). Contents of each worksheet are as follows: worksheet 1) notes for summary tables; worksheet 2) Primers for qPCR validation of CSC50 genes; worksheet 3) table includes values represented in the *Figure 1E* where values are fold changes relative to HL or LH, as indicated; worksheet 4) NY8 and NY15 Bru-seq data related to *Figure 7A*; worksheet 5) NY8 and NY15 ChIP-seq data related to *Figure 7B* with information regarding enhancer binding; worksheet 6) summary of selected genes in expression vs survival in TCGA PAAD tumors - related to *Figure 7E*, S9A, and S9B, worksheet 7) HNF1A-upregulated and -bound genes: association between gene expression and survival in TCGA PAAD tumors - related to *Figure 7E*; worksheet 8) HNF1A-upregulated genes: association between gene expression and survival in TCGA PAAD tumors - related to *Figure 7—figure supplement 1A*; worksheet 9) HNF1A-downregulated genes: association between gene expression and survival in TCGA PAAD tumors - related to *Figure 7—figure supplement 1B*; worksheet 10) TCGA donors used in survival analysis - related to *Figure 7E*, *Figure 7—figure supplement 1A and B*; worksheet 11) Overrepresented TF-binding motifs in cancer stem cell gene set (CSC50), oPOSSUM3 results - related to *Figure 1*; worksheet 12) predicted HNF1A targets - related to *Figure 1*; worksheet 13) overrepresented TF-binding motifs in HNF1A upregulated genes, oPOSSUM3 results - related to *Figure 7D*; worksheet 14) overrepresented TF-binding motifs in HNF1A downregulated genes, oPOSSUM3 results - related to *Figure 7D*; worksheet 15) predicted POU5F1 targets - related to *Figure 7D*; worksheet 16) HNF1A ChIP-seq peak enhancer overlap, NY15 replicate 1 (rep1) - related to *Figure 7B*; worksheet 17) A ChIP-seq peak enhancer overlap, NY15 replicate 2 (rep2) - related to *Figure 7B*; worksheet 18) HNF1A ChIP-seq peak enhancer overlap, NY8 replicate 1 (rep1) - related to *Figure 7B*; worksheet 19) HNF1A ChIP-seq peak enhancer overlap, NY8 replicate 2 (rep2) - related to *Figure 7B*.

DOI: https://doi.org/10.7554/eLife.33947.031

• Transparent reporting form

DOI: https://doi.org/10.7554/eLife.33947.032

## Data availability

All data from this study is available without limitations (GSE108151).

The following dataset was generated:

| Author(s) | Year | Dataset title | Dataset URL | Database, license, and accessibility information |
|---|---|---|---|---|
| Abel E, Goto M, Magnuson B, Abraham S, Ramanathan N, Hotaling E, Alaniz AA, Kumar-Sinha C, Dziubinski ML, Urs S, Wang L, Shi J, Waghray M, Ljungman M, Crawford HC, Simeone DM | 2018 | HNF1A is a Novel Oncogene and Central Regulator of Pancreatic Cancer Stem Cells | https://www.ncbi.nlm.nih.gov/geo/query/acc.cgi?acc=GSE108151 | Publicly available at the NCBI Gene Expression Omnibus (accession no. GSE108151). |

The following previously published dataset was used:

| Author(s) | Year | Dataset title | Dataset URL | Database, license, and accessibility information |
|---|---|---|---|---|
| Broad Institute TCGA Genome Data Analysis Center | 2016 | Analysis-ready standardized TCGA data from Broad GDAC Firehose 2016_01_28 run | http://gdac.broadinstitute.org/runs/stddata__2016_01_28/ | No restrictions; all data available without limitations |

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
