## [Decision Letter]

Thank you for submitting your article "HNF1A is a Novel Oncogene and Central Regulator of Pancreatic Cancer Stem Cells" for consideration by *eLife*. Your article has been reviewed by three peer reviewers, and the evaluation has been overseen by a Reviewing Editor and Kevin Struhl as the Senior Editor. The following individual involved in review of your submission has agreed to reveal their identity: Francisco X Real (Reviewer #2).

The reviewers have discussed the reviews with one another and the Reviewing Editor has drafted this letter to clarify our concerns about this work.

In this manuscript, Abel et al. describe a function for HNF1A as a potential new marker of the pancreatic cancer stem cell, and provide evidence of its oncogenic behavior. They utilize a series of patient derived cell lines and in vitro assays to demonstrate its function. The authors also provide evidence that HNF1A promotes stemness features in PDA through regulation of POU5F1/OCT4. The paper presents interesting new data, providing evidence for the oncogenic role of HNF1A in pancreatic cancer including functions in enhancing tumor initiating cell activity. However, there are some very significant concerns about the inconsistencies of cell lines for given experiments, lack of statistical analyses, and the interpretation of whether HNF1A is truly a CSC marker vs. simply an oncogene. These concerns raise questions as to the robustness of the proposed model.

Major points:

1) Inconsistency in which cell lines were used:

The study uses multiple cell lines (i.e. NY5, NY8, NY15, NY90). While it is useful to have multiple cell lines, in many cases, they seem to be chosen at random and mainly to support the authors hypothesis. The significantly weakens the study. Some examples of this include:

A) In Figure 2A, there looks to be little to no expression of HNF1A in NY8 or NY15 and yet surprisingly in Figure 2B, they then do knockdowns in those cell lines which in the previous Western blot looked to barely have any expression. If NY8 has no HNF1A expression, then how did they statistically quantify the effect of knockdown?

B) Using siRNA against HNF1A (Figure 2C), the authors have shown that HNF1A-depletion strongly attenuates the growth of primary PDA cells NY5, NY8 and NY15, and produces a more modest effect on NY90. The P2 population is also reduced following HNF1A-depletion in these lines (Figure 3A), but present no statistics to support this. The authors suggest that basal HNF1A levels may predict dependency, but it is not clear whether the NY90 line has lower HNF1A protein levels than the other lines. Also, it would seem that correlating HNF1A levels specifically in the P2 population with dependency would be more consistent with the authors proposed model that HNF1A has specific functions in this sub-population of cells. These questions could be clarified by associating response to HNF1A knockdown and expression in the P2 population and by adding additional lines to the analysis.

C) In Figure 4, the authors again use the NY8 line, which shows essentially no HNF1A expression, yet that is the line they previously (in Figure 3E) used to support knockdown experiments and decreased tumorsphere formation. How do they explain this discrepancy?

D) In Figure 1—figure supplement 1, they use NY8 for Figure 1—figure supplement 1C but then NY15 for Figure 1—figure supplement 1D. Similarly, for Figure 1—figure supplement 2, they use NY15 for Figure 1—figure supplement 2D, then unclear what they used for Figure 1—figure supplement 2E.

These inconsistencies make the manuscript difficult to follow. To address this, the authors need to provide consistent data across the cell lines for each assay they have done. If a particular cell line is not appropriate, they need to explain the rationale for choosing that particular line. Moreover, it is essential that they provide statistical analysis of each assay performed.

2) HNF1A as a CSC marker versus an oncogene

It is clear from the data that HNF1A promotes the growth of PDAC cells, and thus is likely an important oncogene. However, what is less clear is whether this is truly a CSC marker versus just an oncogene promoting growth. In Figure 1C, it does not look like there is much enrichment for HNF1A in NY8 cells, and in fact the co-expression of CD44 appears lower in the P2 fraction. In Figure 1D, the DPP fraction, which they argue is a HNF1A target, does not appear especially enriched in the P2 population in the NY5 line. Along the same lines, in Figure 1—figure supplement 1, there does not appear to be any enrichment of HNF1A in the stem cell fraction in NY5 cells. In Figure 3E, couldn't the decreased tumorspheres simply be due to a lower rate of proliferation as seen in Figure 2? This does not seem to support a stem cell function as much as a proliferation function. In Figure 4C, the authors once again switch to NY15 cells for unclear reasons, but the bigger issue with Figure 4, however, is that this is not really arguing for a CSC function – it is mainly arguing that HNF1A acts as a classical oncogene – it leads to more colony formation (Figure 4F and 4G) and cooperates with KRAS (Figure 4H and 4I).

Along the same lines, in Figure 5, the authors use mouse xenografts to argue that HNF1A is a CSC marker with knockdown studies. But the assays they did in Figure 5A-C really just argues it is an oncogene. To show stem cell function, they would need to do serial transplantation and limiting dilution analysis. I am not sure exactly what the data in Figure 5C is supposed to show, since the subcutaneous tumors are not very physiologically relevant yet this is where they show the depletion of the EPCAM/CD44/CD24 population – why wasn't this done in the pancreatic tumors instead? Also, in Figure 5D, there does not appear to be much knockdown in HNF1A#2, so why do they then see this quantification in NY5 cells in Figure 5E? In Figure 6D, the authors present data that knockdown of either HNF1A or OCT4 decreases tumorsphere, but along the same lines as previously noted, are these cells just sicker and/or not proliferating as well? This is especially true for the knockdown of OCT4 – how does this affect growth or apoptosis? The photo shown in Figure 6G would suggest that the OCT4 knockdown cells are not very healthy. One way of addressing this issue is via the use of inducible shRNAs and/or cDNA rescue, to show that you can discern the growth inhibition effects versus just sick cells.

3) HNF1A/OCT4

It is important to show the data on the ChIP-PCR of HNF1A at the OCT4 promoter (mentioned in the Discussion) and provide some more detail about the putative elements involved in regulation. It would be desirable to show promoter-reporter assays at this locus to demonstrate this is really important in its function.

4) Bioinformatics issues

A more detailed description of the ChIP-seq methods (description of experiments, replicates, number of peaks, concordance of replicates, etc.) and analysis of the results of these experiments. In addition, a more detailed description of the merged results of RNA-Seq and ChIP-seq would be valuable, including motif analysis of the promoter of genes whose expression was deregulated upon HNF1A knockdown (is OCT4 motif enriched? Any other candidates? Are OCT4 targets enriched as well among the deregulated genes?). A more detailed study of the possible role of distant regulatory sites through the analysis of ENCODE data for enhancer activity would be also important to support the regulatory role on target genes.

Regarding the analysis of association with prognosis: were these analyses corrected for multiple testing? Were signatures better than individual genes? Could the data be validated in the ICGC series? These are relevant questions that should be readily addressable.

Can the authors clarify whether OCT4 expression correlates with HNF1A expression across PDA cell lines and tumors and whether OCT4 is enriched in the P2 population?

---

## [Author Response]

Major points:1) Inconsistency in which cell lines were used:The study uses multiple cell lines (i.e. NY5, NY8, NY15, NY90). While it is useful to have multiple cell lines, in many cases, they seem to be chosen at random and mainly to support the authors hypothesis. The significantly weakens the study. Some examples of this include:A) In Figure 2A, there looks to be little to no expression of HNF1A in NY8 or NY15 and yet surprisingly in Figure 2B, they then do knockdowns in those cell lines which in the previous Western blot looked to barely have any expression. If NY8 has no HNF1A expression, then how did they statistically quantify the effect of knockdown?B) Using siRNA against HNF1A (Figure 2C), the authors have shown that HNF1A-depletion strongly attenuates the growth of primary PDA cells NY5, NY8 and NY15, and produces a more modest effect on NY90. The P2 population is also reduced following HNF1A-depletion in these lines (Figure 3A), but present no statistics to support this. The authors suggest that basal HNF1A levels may predict dependency, but it is not clear whether the NY90 line has lower HNF1A protein levels than the other lines. Also, it would seem that correlating HNF1A levels specifically in the P2 population with dependency would be more consistent with the authors proposed model that HNF1A has specific functions in this sub-population of cells. These questions could be clarified by associating response to HNF1A knockdown and expression in the P2 population and by adding additional lines to the analysis.

We agree that the levels of HNF1A in the cell lines used might have been confusing in the original submission, and have re-configured the figures as well as added numerous additional new western blots to demonstrate expression data in a more robust way. We have focused our studies on 3 low-passage primary human PDA lines (NY5, NY8, NY15), and have used them throughout the manuscript to maintain consistency. Additionally, quantitation of Western blots has been added throughout the manuscript to help with interpretation, including that HNF1A protein is elevated in the P2 (EPCAM+/CD44+) subpopulation and that HNF1A expression is tightly linked to PCSC marker expression.

C) In Figure 4, the authors again use the NY8 line, which shows essentially no HNF1A expression, yet that is the line they previously (in Figure 3E) used to support knockdown experiments and decreased tumorsphere formation. How do they explain this discrepancy?

We have included clear data by western blotting (Figure 2A) and mRNA expression (Figure 2—figure supplement 1A) that demonstrates expression of HNF1A in NY8 cells. Further, we have demonstrated that both knockdown (Figure 3C-E), and overexpression of HNF1A (Figure 4B and 4E) is significantly correlated with tumorsphere formation.

D) In Figure 1—figure supplement 1, they use NY8 for Figure 1—figure supplement 1C but then NY15 for Figure 1—figure supplement 1D. Similarly, for Figure 1—figure supplement 2, they use NY15 for Figure 1—figure supplement 2D, then unclear what they used for Figure 1—figure supplement 2E.These inconsistencies make the manuscript difficult to follow. To address this, the authors need to provide consistent data across the cell lines for each assay they have done. If a particular cell line is not appropriate, they need to explain the rationale for choosing that particular line. Moreover, it is essential that they provide statistical analysis of each assay performed.

As stated above, we have streamlined our use of consistent primary PDA cell lines throughout the manuscript. In addition, we have provided statistical analysis for each assay performed throughout the manuscript.

2) HNF1A as a CSC marker versus an oncogeneIt is clear from the data that HNF1A promotes the growth of PDAC cells, and thus is likely an important oncogene. However, what is less clear is whether this is truly a CSC marker versus just an oncogene promoting growth. In Figure 1C, it does not look like there is much enrichment for HNF1A in NY8 cells, and in fact the co-expression of CD44 appears lower in the P2 fraction.

We provide clear data throughout the manuscript that HNF1A is upregulated in NY8 cells in the P2 population and that this population possesses CSC functionality, including self-renewal, expression of cancer stem cell markers, sphere formation, and increased tumor initiation. In addition to the differentially expressed HNF1A mRNA quantitated in Figure 1C, quantitation of HNF1A protein levels from sorted NY5, NY8, and NY15 has been added to clarify the association of HNF1A expression and the P2 subpopulation. Cells in the P2 subpopulation express medium levels of EPCAM and CD44 when compared to the P1 and P3 (Figure 1—figure supplement 1B), and as such we have re-described the P2 cells as “CD44^Med^/EPCAM^Med^.”

In Figure 1D, the DPP fraction, which they argue is a HNF1A target, does not appear especially enriched in the P2 population in the NY5 line. Along the same lines, in Figure 1—figure supplement 1, there does not appear to be any enrichment of HNF1A in the stem cell fraction in NY5 cells.

To address the concerns regarding expression of DPP4, an additional known transcriptional target of HNF1A, we have included quantitated Western blots to support that DPP4 is most highly expressed in the P2 subpopulation (Figure 1—figure supplement 1B), similar to HNF1A and CDH17.

In Figure 3E, couldn't the decreased tumorspheres simply be due to a lower rate of proliferation as seen in Figure 2? This does not seem to support a stem cell function as much as a proliferation function.

Tumorsphere formation is a well-established in vitro measure of CSC function. In Figures 1C and 1D we show that the P1 and P3 subpopulations of PDA cells, which have reduced HNF1A expression, are also markedly reduced in their ability to form tumorspheres without any additional perturbation (i.e. HNF1A knockdown). Additionally, HNF1A expression is increased in tumorspheres (Figure 1—figure supplement 2). In Figures 3 and 4, we demonstrate that altering HNF1A expression in PDA cells results in increased or decreased tumorsphere forming activity in a manner that corresponds to HNF1A expression levels. Collectively, these data support a tight association between HNF1A levels and tumorsphere formation. Lastly, Figure 6 and Figure 6—figure supplement 2 demonstrate that OCT4 is the mechanistic link between HNF1A and sphere formation, and importantly that knockdown of OCT4 does not affect cell cycle or apoptosis in PDA cells (Figure 6—figure supplement 2).

In Figure 4C the authors once again switch to NY15 cells for unclear reasons, but the bigger issue with Figure 4, however, is that this is not really arguing for a CSC function – it is mainly arguing that HNF1A acts as a classical oncogene – it leads to more colony formation (Figure 4F and 4G) and cooperates with KRAS (Figure 4H and 4I).

The updated Figure 4 now contains overexpression of HNF1A in three primary PDA models (NY8, NY15, and NY53) throughout the figure. The purpose of using these cells, which was to test whether overexpression of HNF1A could further promote CSC-marker expression and tumorsphere formation in PDA cells with varying endogenous levels of HNF1A, is now clearly stated in the Results. The results show that HNF1A overexpression increases both CSC-marker expression and tumorsphere formation, consistent with the knockdown data in Figure 3.

We agree with the reviewer that HNF1A is capable of acting as an oncogene and have chosen this suggested designation throughout the revised manuscript, including the title. That said, our data also support that part of its oncogenic activity is through its regulation of PCSC function like other oncogenes we and others have identified, such as NOTCH (Abel et al., 2014), BMI1 (Proctor et al., 2013), c-MET (Li et al., 2011a; Li et al., 2011b), and NRAS (Li et al., 2013).

Along the same lines, in Figure 5, the authors use mouse xenografts to argue that HNF1A is a CSC marker with knockdown studies. But the assays they did in Figure 5A-C really just argues it is an oncogene. To show stem cell function, they would need to do serial transplantation and limiting dilution analysis.

We agree with the reviewer that limiting dilution analysis is a valuable assay to assess CSC function. Due to the time constraints imposed for revision, this assay was not feasible. However, to acknowledge these concerns, we have adopted the language that HNF1A acts as an oncogene that regulates PSCS properties. To support the latter claim, we demonstrate increased expression of HNF1A in a CSC population capable of self-renewal and generation of the diverse progeny observed in the original tumor sample (Figure 1 and Figure 1—figure supplement 1), enhanced expression of HNF1A in tumor-sphere forming conditions (Figure 1—figure supplement 2), the ability of HNF1A to regulation expression of CSC markers (Figures 3 and 4; and related figure supplements) and enhancement of tumor growth (Figure 5 and Figure 5—figure supplement 1). In the revised manuscript, we also provide new data (Figure 6; Figures 6—figure supplements 1 and 2) that HNF1A is linked to the stem cell protein OCT4 as a critical downstream mediator of HNF1A function, through its direct interaction of its distal promoter.

We conclude that HNF1A joins a group of oncoproteins that regulate cancer stem cell activity (e.g. c-MET, NOTCH, BMI1, etc.).

I am not sure exactly what the data in Figure 5C is supposed to show, since the subq tumors are not very physiologically relevant yet this is where they show the depletion of the EPCAM/CD44/CD24 population – why wasn't this done in the pancreatic tumors instead?

We have previously shown that assays assessing PCSC function are similar in head-to-head orthotopic and subcutaneous implantation experiments (Li et al., 2007; Li et al., 2011a), demonstrating that the subcutaneous model is equally effective as the orthotopic model in defining PCSC function. Consistent with our previous findings, both orthotopic and subcutaneous tumor assays had similar results with regards to tumor growth in this study. Additionally, orthotopic tumors were labeled with GFP-luciferase, which prevented use of flow cytometry to quantitate PCSCs. Unlabeled cells were used for the subcutaneous tumor models, which allowed for PCSC staining. The results of these two experiments are complementary.

Also, in Figure 5D there does not appear to be much knockdown in HNF1A#2, so why do they then see this quantification in NY5 cells in Figure 5E?

We have provided a clearer representative image in Figure 5D to demonstrate that the HNF1A shRNA#2 provide significant knockdown effects, and the cumulative results in Figure 5E show the effectiveness of HNF1A knockdown in depleting the CSC population in vivo(p<0.001 for both shRNAs).

In Figure 6D, the authors present data that knockdown of either HNF1A or OCT4 decreases tumorsphere, but along the same lines as previously noted, are these cells just sicker and/or not proliferating as well? This is especially true for the knockdown of OCT4 – how does this affect growth or apoptosis? The photo shown in Figure 6G would suggest that the OCT4 knockdown cells are not very healthy. One way of addressing this issue is via the use of inducible shRNAs and/or cDNA rescue, to show that you can discern the growth inhibition effects versus just sick cells.

We have added new data to show that PDA cells are responsive to OCT4 knockdown in tumorsphere assays (Figure 6E and 6I), and did not show a significant change in cell cycle (propidium iodide staining) or apoptosis (annexin V staining) following knockdown of OCT4 (Figure 6—figure supplements 2A and 2B). These data indicate that knockdown of OCT4 is not preventing tumorsphere formation via cell death or cytostasis in PDA cells. Additionally, OCT4 is a well-established stem cell regulator (Okita et al., 2007; Takahashi and Yamanaka, 2006), and has been shown to play a role in cancer stem cells as well (Lu et al., 2013; Kumar et al., 2012; Nishi et al., 2013). Our data is consistent with the canonical role of OCT4 as a key regulator of stemness, both in normal cells and in cancer.

3) HNF1A/OCT4It is important to show the data on the ChIP-PCR of HNF1A at the OCT4 promoter (mentioned in the Discussion) and provide some more detail about the putative elements involved in regulation. It would be desirable to show promoter-reporter assays at this locus to demonstrate this is really important in its function.

We appreciate this comment and have in fact done several additional experiments to support that HNF1A binds to the OCT4 promoter. Using ChIP-PCR, we were able to demonstrate enrichment of the OCT4 LTR promoter (Malakootian et al., 2017) (Figure 6—figure supplement 1C), previously mentioned in the Discussion. Additionally, we were able to demonstrate that this region functions as an HNF1A-responsive promoter in promoter-reporter assays (Figure 6—figure supplement 1D). These data support a direct regulatory interaction between HNF1A and the OCT4 LTR promoter.

4) Bioinformatics issuesA more detailed description of the ChIP-seq methods (description of experiments, replicates, number of peaks, concordance of replicates, etc.) and analysis of the results of these experiments.

We have added more detail about the Bru-seq/ChIP-seq in the Results and Materials and methods section.

In addition, a more detailed description of the merged results of RNA-Seq and ChIP-seq would be valuable, including motif analysis of the promoter of genes whose expression was deregulated upon HNF1A knockdown (is OCT4 motif enriched? Any other candidates? Are OCT4 targets enriched as well among the deregulated genes?).

We used an overrepresentation analysis to find TF binding sites in promoters of HNF1A-responsive genes (see Figure 7D and Supplementary file 2 for the complete results). POU5F1 (OCT4) was the most highly ranked predicted TF for the HNF1A-repressed genes. OCT4 targets were indeed found among the HNF1A-repressed genes (now included in the Results).

A more detailed study of the possible role of distant regulatory sites through the analysis of ENCODE data for enhancer activity would be also important to support the regulatory role on target genes.

Using ENCODE enhancer data from 6 merged cell lines we examined ChIP-seq peak overlap with identified enhancer regions. For the HNF1A-bound genes represented in Figure 7B, we indicated the fraction of genes with enhancer-overlapping peaks. We have also added a list of merged enhancer regions and a list of peaks indicating overlap or non-overlap in the Supplementary file 2. 72.7% of HNF1A-bound genes had peaks overlapping in at least one of these putative enhancer regions, suggesting that HNF1A has significant interaction with regulatory regions.

Regarding the analysis of association with prognosis: were these analyses corrected for multiple testing?

The original analysis presented did not apply p value adjustment before selecting significant genes. In the process of revisiting this issue, we decided to use a Cox PH model for survival, since gene expression is a continuous variable and is thus well suited, instead of the median-stratification model we used previously. We present the revised analysis showing both the FDR 0.1 and 0.25 as thresholds for gene selection (Figure 7, Figure 7—figure supplement 1). The permutation tests at the different thresholds agree, suggesting the selection threshold is not critical.

Were signatures better than individual genes?

We have added additional analysis of the TCGA dataset. Permutation tests showed that HNF1A-activated genes were significantly associated with poorer outcomes versus randomly selected genes (insets, Figure 7E and Figure 7—figure supplement 1A; see Materials and methods for details). These findings further support the oncogenic role for HNF1A in PDA as a direct regulator of a set of genes associated with poor patient survival.

Could the data be validated in the ICGC series? These are relevant questions that should be readily addressable.

As suggested, we applied our survival analysis to the ICGC data set (PACA-AU cohort). For HNF1A activated/bound genes, 7 overlapping genes were identified in this manner using both the TCGA and ICGC analyses. The ICGC result contrasts the TCGA result in that genes associated with increased survival were also identified, whereas in the TCGA result, no such genes were found (Author response image 1). A likely source of this discrepancy is a general disagreement among the single-gene survival models. At p < 0.01 (highly significant) very few background genes overlapped between the two datasets (Author response image 2).

It is possible that the underlying difference between the ICGC and TCGA datasets is the relatively high tumor cellularity requirement for ICGC samples (at least 40%) (Waddell et al., 2015; Bailey et al., 2016). By contrast, TCGA also utilized samples with low tumor cellularity (Network, 2017), a common hallmark of PDA. While it is unclear that sample selection criterion accounts for this disparity between datasets, we do however believe that the usage of a more representative collection of samples by the TCGA justifies our reliance on the dataset in our study. We are receptive to alternative solutions should the reviewer disagree.

**Author response image 1. respfig1:** Comparison of TCGA and ICGC survival analyses.

**Author response image 2. respfig2:** Comparison of TCGA and ICGC background gene concordance. Overlap of all genes is on the left, “UP” refers to increased survival genes, “DN” to reduced survival genes.

Can the authors clarify whether OCT4 expression correlates with HNF1A expression across PDA cell lines and tumors and whether OCT4 is enriched in the P2 population?

Using qRT-PCR analysis to quantitate both HNF1A and POU5F1 (OCT4) mRNA levels in 22 primary PDA cell lines as well as HPNE and HPDE cells, we found a significant positive correlation (r=0.5189, p=0.0094, Figure 6C) between the mRNA levels of both genes. Additionally, we also found a positive correlation (r=0.4064, p=8.9x10^-8^, Author response image 3) between HNF1A and POU5F1 mRNA levels in patient data from TCGA. We did not, however, observe a significant association between POU5F1 mRNA and any of the PDA subpopulations (Author response image 3). This would suggest that factors other than HNF1A modulate the levels of POU5F1 mRNA in different PDA subpopulations.

**Author response image 3. respfig3:** Correlation of HNF1A and POU5F1 expression.